# Meta-AdaM: A Meta-Learned Adaptive Optimizer with Momentum for Few-Shot Learning

**Siyuan Sun**
Department of Computer Science
Iowa State University
Ames, IA 50011
sxs14473@iastate.edu

**Hongyang Gao**
Department of Computer Science
Iowa State University
Ames, IA 50011
hygao@iastate.edu

## Abstract

We introduce Meta-AdaM, a meta-learned adaptive optimizer with momentum, designed for few-shot learning tasks that pose significant challenges to deep learning models due to the limited number of labeled examples. Meta-learning has been successfully employed to address these challenges by transferring meta-learned prior knowledge to new tasks. Most existing works focus on meta-learning an optimal model initialization or an adaptive learning rate learner for rapid convergence. However, these approaches either neglect to consider weight-update history for the adaptive learning rate learner or fail to effectively integrate momentum for fast convergence, as seen in many-shot learning settings. To tackle these limitations, we propose a meta-learned learning rate learner that utilizes weight-update history as input to predict more appropriate learning rates for rapid convergence. Furthermore, for the first time, our approach incorporates momentum into the optimization process of few-shot learning via a double look-ahead mechanism, enabling rapid convergence similar to many-shot settings. Extensive experimental results on benchmark datasets demonstrate the effectiveness of the proposed Meta-AdaM.

## 1 Introduction

Deep learning has demonstrated its capability in solving various challenging tasks in many-shot-learning settings, where each classification class contains sufficient training examples [12]. However, its performance is hindered on small datasets or under few-shot-learning (FSL) settings [37]. In FSL, unseen tasks with limited training examples pose significant challenges for optimizing deep learning models [45]. With limited training data, deep learning models may suffer from severe underfitting or overfitting problems [28]. Numerous meta-learning methods have been proposed to address this challenge by identifying and encoding shared knowledge among different data distributions within the model. With prior generalized knowledge, a meta-learning model can be rapidly transferred to a new task by fine-tuning it with limited unseen data [13].

Existing optimization-based meta-learning methods for FSL facilitate this transfer process by fine-tuning a meta-learned model on new few-shot tasks. There are two primary directions for using meta-learning in FSL tasks: learning an optimal initialization and generating high-quality gradient updates for rapid convergence. In the first direction, most methods [7, 8, 29, 24, 17, 18] employ an inner loop to fine-tune the meta-model on specific tasks and iteratively optimize the meta-model in an outer loop based on the adapted model from the inner loop. The resulting initial model weights enable the adapted model to converge more quickly to an optimal solution for a new task. In the second direction, some works [38, 2, 46] focus on generating high-quality and task-adaptive gradient updates, allowing the optimizer to rapidly converge to an appropriate solution.

37th Conference on Neural Information Processing Systems (NeurIPS 2023).

Our work focuses on the second direction. Most existing works in this direction aim to use adaptive learning rates by considering weights and gradients [18, 30, 3]. However, it has been shown that the weight-update history is more important than the weights themselves for determining an appropriate learning rate. Additionally, existing works often fail to incorporate momentum in FSL, which has been shown to be effective in accelerating the optimization process in many-shot settings. This is due to the fact that momentum typically exhibits high fluctuations during the initial updates [19]. Another challenge for the meta-trained base model is the incorporation of prior knowledge. Certain classes exhibit features that align more closely with prior knowledge than others, leading to an imbalance in predictions generated by the model.

To address these challenges, we propose a meta-learned adaptive optimizer with momentum (Meta-AdaM). To learn a more accurate adaptive learning rate, our approach meta-learns a learning rate learner based on gradient update history instead of weights. The learned learning rate learner, along with the meta-learned model initialization, is applied when adapting to new tasks. Moreover, we introduce a dynamic class weight scheme for classification tasks, aimed at effectively balancing the loss distribution among various classes, which can help the model generate unbiased results.

## 2 Related Works

In this section, we introduce related works to our proposed approach.

### 2.1 Few-Shot Learning and Meta Learning

A formal definition of machine learning can be described as follows: a computer program can learn some experience $E$ from a set of classes that belongs to task $T$ with performance metric $P$ if $P$ increases with $E$ on $T$ measured by $P$ [22]. Few-shot-learning problems are a subset of machine learning problems in which $E$ only contains a limited number of samples [45]. Under this setting, training deep neural networks can be challenging. The empirical risk $R$ generated from the limited experience $E$ can differ greatly from the expected empirical risk. Meta-learning enhances the performance $P$ by learning shared features from other data. The metric-based [36] and model-based meta-learning methods [35, 23, 21] rely on extra features or models to improve the few-shot learning capabilities. Recently, optimization-based meta-learning methods have obtained more attention for their strong generalization ability. The optimization-based methods reduce the meta-Learning problem into a bi-level optimization problem. The inner loop optimizes the base model on a certain task, and the outer loop optimizes the base model across several tasks to adjust the initial weight for quick adaption. Without introducing new elements, such a structure has the potential to adapt better to unseen data. The most representative optimization-based method is the MAML method [7]. Subsequent MAML variants [8, 29, 24, 17, 18, 34, 9] focus on optimizing the optimization process.

### 2.2 Gradient-Descent Based Optimizer

The optimizer plays a crucial role in enhancing the efficiency of the training process in deep learning. Most popular optimizers are based on the gradient descent algorithm [1]. To further improve training efficiency and increase convergence speed, existing optimizers, such as AdaM [14], often utilize the first and second orders of momentum. The first order of momentum, also known as momentum, is the accumulated sum of historical gradients during the training process, which can be considered a more effective update direction compared to gradients. The second order of momentum is used to control the step size for each parameter by estimating the update velocity for each parameter. Moreover, based on AdaM, other optimizers [14, 6, 19] introduce additional elements to better estimate the update direction and step size. Learned optimizers [2, 1, 43] utilize a trainable model to estimate the update direction and step size, which can better adapt to new tasks.

#### 2.2.1 Optimizer for Few-Shot Learning

Meta-learning has shown a strong ability to solve few-shot learning problems. So, researchers also use meta-learning methods to train optimizers for few-shot learning problems. [18] propose an optimizer that works as the stochastic gradient descent. [30, 3] propose to use linear models to carefully estimate the learning rate for each parameter. [25] consider estimating curvature instead of gradients. However, these works only consider the gradient change in one step and ignore the

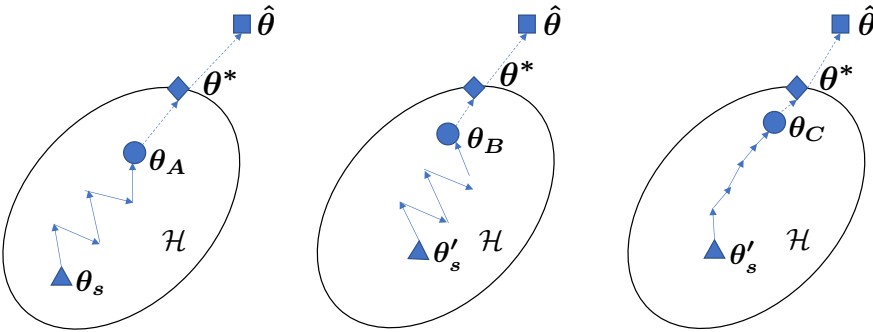

A. Regular initilization with random optimization

B. Meta-learned initialization

C. Meta-learned Optimizer With Meta-learned Initialization

Figure 1: Illustration of different few-shot-learning optimizing strategies. $\hat{\theta}$ is the global optimum solution. $\theta^*$ is the best approximation in the hypothesis space $\mathcal{H}$ determined by the FSL model. (A) shows the optimization trajectory of naive optimizer with random initialization $\theta_s$ and regular weights updates, leading to a poor solution $\theta_A$. (B) illustrates using a regular weights update with meta-learned initialization $\theta'_s$, resulting in a better approximation $\theta_B$ than $\theta_A$. In (C), a meta-learned optimizer is used with the meta-learned initialization $\theta'_s$, leading to the best approximation $\theta_C$.

gradient change history in the fine-tuning process, which is not optimal. In this work, we propose an optimizer called Meta-AdaM, specially designed for the inner loop in meta-learning. We design an LSTM-based meta-learner to adaptively estimate the learning rate. Also, we make wiser use of momentum to dynamically determine the optimal updating direction. This can help the base model to converge to the optimum in fewer steps.

# 3 Meta-AdaM: A Meta-Learned Adaptive Optimizer with Momentum

We propose a **Meta**-learned **Ada**ptive optimizer with **M**omentum (**Meta-AdaM**) for FSL.

## 3.1 Motivations and Challenges

Few-shot learning (FSL) is a formidable problem in machine learning, where models are expected to generalize effectively to new tasks with only a handful of labeled examples. The limited amount of training data, however, poses a significant challenge for traditional optimizers, as they tend to overfit and yield subpar solutions [28]. As demonstrated in [45], the empirical risk minimizer or optimizer in FSL is unreliable due to its propensity to overfit a limited number of training examples, resulting in unsatisfactory solutions, as illustrated in Figure 1 (A). Consequently, optimization steps are often restricted to prevent overfitting, which makes it difficult for an optimizer to discover suitable solutions for FSL tasks. Generally, two primary strategies exist to tackle this issue. One involves identifying an optimal initialization for model parameters, as depicted in Figure 1 (B). The other focuses on generating high-quality gradient updates for rapid convergence to an appropriate solution, as displayed in Figure 1 (C). These strategies are orthogonal and can be combined. The objective of this work is to develop an optimizer capable of producing high-quality gradient updates, allowing the model to achieve a satisfactory optimum with a limited number of updates.

The most commonly used optimization algorithms for training deep learning models are variants of gradient descent, which utilize the following update formula:

$$\theta_i^t = \theta_i^{t-1} - \eta_i^t \times (\alpha_i^t \times g_i^t + \beta_i^t \times m_i^{t-1}), \tag{1}$$

$$m_i^t = \alpha_i^t \times g_i^t + \beta_i^t \times m_i^{t-1}. \tag{2}$$

In the update formula, $\theta_i^t$ represents the value of a trainable parameter $\theta_i$ of the model after undergoing $t$ updates. The variables $\eta_i^t$, $g_i^t$, and $m_i^t$ refer to the learning rate, gradient, and first-order momentum for $\theta_i$ at the $t^{th}$ update, respectively. Additionally, $\alpha_i^t$ and $\beta_i^t$ are coefficients to balance the gradients and momentum, where momentum is an exponentially weighted moving average of the gradients.

In many-shot learning scenarios, where each learning task has abundant labeled data, several optimizers [41, 14, 32], such as Adam [14] and SGD with momentum [27], have been developed to speed up

the convergence speed by incorporating adaptive learning rates [5] and momentum. However, these optimization algorithms cannot be directly applied to FSL tasks as the variance of momentum in the initial optimization steps tends to be high [19], leading to unstable gradient updates with limited optimization steps and resulting in poor local optima.

Meta-learning has recently demonstrated a robust capacity to address FSL problems. Specifically, a meta-learner is trained on similar tasks and employed to learn a learner for any new task [38, 2, 46]. While several approaches [18, 30, 3] have been proposed to meta-learn adaptive learning rates in FSL by incorporating gradients and weights, the relationship between learning rates and weight-changing history, rather than just weight values, has been shown to be crucial [11, 15]. Furthermore, effectively utilizing momentum has been challenging due to its high variance in the initial updates [19]. In light of these challenges, we propose a novel approach, called Meta-AdaM, which meta-learns an adaptive optimizer with momentum. Our approach leverages the power of meta-learning to learn a model that predicts adaptive learning rates based on gradient update history, and effectively integrates momentum to speed up the convergence speed. By doing so, Meta-AdaM enables a learner to quickly adapt to new few-shot learning tasks with only a limited number of update steps.

## 3.2 Update History-Based Adaptive Learning Rates through Meta-Learning

In this section, we propose utilizing meta-learning to develop a learner that generates adaptive learning rates based on gradient update history. Existing methods, such as [30, 3], depend on a meta-trained multi-layer perceptron to predict learning rates using model parameter values and gradients. However, recent studies indicate that the relationship between learning rates and weight-changing history is more critical than between learning rates and weight values [11, 15]. As a result, it is essential to predict adaptive learning rates based on weight-changing history, specifically gradient $g_i^t$ and momentum $m^{t-1}i$. To achieve this, we treat the trajectory of $g_i^t$ and $m_i^t$ as sequence data and employ a long short-term memory network (LSTM) [39]. The LSTM accepts momentum and new gradients as inputs and outputs an adaptive learning rate for each trainable parameter $\theta_i$. Formally, the update formula for $\theta_i$ with the proposed meta-learner is defined as follows:

---
**Algorithm 1** MetaMomentumInner

1: **Input:** $\boldsymbol{\theta}$: meta model, learning rate learner: LSTM, $S_{\mathcal{T}}$: data of a task.
2: **Require:** $\eta$: base step size, $K$: number of inner loops, $f(\cdot)$: loss function.
3: Set $\boldsymbol{\theta}^0 = \boldsymbol{\theta}$, $\boldsymbol{m}^0 = \boldsymbol{0}$
4: **for all** $t = 1$ to $K$ **do**
5:     $\boldsymbol{g}^t = \nabla_{\boldsymbol{\theta}^{t-1}} f(S_{\mathcal{T}}, \boldsymbol{\theta}^{t-1})$
6:     $\boldsymbol{\theta}_m^t = \boldsymbol{\theta}^{t-1} - \eta \boldsymbol{m}^{t-1}$
7:     $\boldsymbol{\theta}_g^t = \boldsymbol{\theta}^{t-1} - \eta \boldsymbol{g}^t$
8:     $\Delta \mathcal{L}_m^t = f\left(S_{\mathcal{T}}, \boldsymbol{\theta}_m^t\right) - f\left(S_{\mathcal{T}}, \boldsymbol{\theta}^{t-1}\right)$
9:     $\Delta \mathcal{L}_g^t = f\left(S_{\mathcal{T}}, \boldsymbol{\theta}_g^t\right) - f\left(S_{\mathcal{T}}, \boldsymbol{\theta}^{t-1}\right)$
10:    $\alpha^t, \beta^t = \sigma(\Delta \mathcal{L}_m^t, \Delta \mathcal{L}_g^t)$
11:    $\boldsymbol{\eta}^t = \text{LSTM}(\alpha^t \boldsymbol{g}^t, \beta^t \boldsymbol{m}^{t-1})$
12:    $\boldsymbol{m}^t = \alpha^t \boldsymbol{g}^t + \beta^t \boldsymbol{m}^{t-1}$
13:    $\boldsymbol{\theta}^t = \boldsymbol{\theta}^{t-1} - \boldsymbol{\eta}^t \times \boldsymbol{m}^t$
14: **end for**
15: **return** $\boldsymbol{\theta}^K$

---

$$\eta_i^t = \text{LSTM}(\alpha_i^t \times g_i^t, \beta_i^t \times m_i^{t-1}), \tag{3}$$

$$\theta_i^t = \theta_i^{t-1} - \eta_i^t \times (\alpha_i^t \times g_i^t + \beta_i^t \times m_i^{t-1}), \tag{4}$$

where $\eta_i^t$ is the predicted learning rate for the trainable parameter $\theta_i$ in the $t^{th}$ update. Compared to previous methods that primarily employ multi-layer perceptrons to predict learning rates based on model parameter values and gradients, our approach naturally considers weight-changing history, leading to more accurate learning rate predictions. For now, we set $\alpha_i^t = 1$ and $\beta_i^t = 0$ temporarily to exclude momentum from the update. This is because the variance of momentum is exceptionally high in the initial update steps, rendering it unsuitable for direct application in FSL tasks. In the next section, we propose an effective method to incorporate momentum, accelerating convergence.

It is crucial to emphasize that our approach fundamentally differs from Meta-LSTM [30]. Meta-LSTM primarily employs a multi-layer perceptron to predict learning rates and weight decay, taking current weights and gradients as input without considering the weight-update history.

## 3.3 Double Lookahead for Effective Momentum Integration

In this section, we propose to effectively integrate momentum in the FSL optimizer to accelerate the convergence speed by using double lookahead. To generate more effective gradients updates during the training process for few-shot-learning tasks, we compute the adaptive model updates by applying lookahead to both momentum and new gradients, leading to a more stable learning

trajectory and better generalization. Lookahead is an effective optimization strategy by looking ahead at a sequence of fast weights [47] for fast converge and a corresponding slow gradient to smooth oscillating. However, in few-shot-learning context, the fast weight will easily trigger the overfitting problem. And the limited update steps number make the slow gradient hard to smooth the oscillating.

We propose addressing this challenge by judiciously selecting the coefficients of the momentum and new gradients using one-step lookahead. Specifically, before performing the accumulation process in Eq. (2), we apply lookahead to both the momentum and the new gradients, resulting in two losses. In this context, the aggregation of weight updates should prioritize the update direction with lower loss. To achieve this, we employ a softmax operator to obtain the coefficients by normalizing the loss decreases. Formally, the proposed approach is defined as in Algorithm 1. In this algorithm, $S_{\mathcal{T}}$ is a batch of support data, $\eta$ is a fixed learning rate, $\sigma(\cdot)$ is a softmax operator, and $f(\cdot)$ is a loss function that computes loss for a batch data $S_{\mathcal{T}}$ with a given model $\boldsymbol{\theta}$. We

---

**Algorithm 2** Meta-AdaM

1: **Input:** Task distribution $p(\mathcal{T})$
2: **Require:** $\alpha, \beta$: step size hyper-parameters
3: Randomly initialize meta model $\boldsymbol{\theta}$
4: Randomly initialize weights in LSTM: $\boldsymbol{\theta}_l$
5: **while** not done **do**
6:      Sample a batch of tasks $\mathcal{T}_i \sim p(\mathcal{T})$
7:      **for all** $\mathcal{T}_i$ **do**
8:          Finetune $\boldsymbol{\theta}$ with $K$ examples:
9:          $\boldsymbol{\theta}_i' = \text{MetaMomentumInner}(\boldsymbol{\theta}, \text{LSTM}, \mathcal{T}_i)$
10:     **end for**
11:     Update meta model and LSTM:
12:     $\boldsymbol{\theta} \leftarrow \boldsymbol{\theta} - \alpha \nabla_{\boldsymbol{\theta}} \sum_{\mathcal{T}_i} \mathcal{L}_{\mathcal{T}_i}(f_{\boldsymbol{\theta}_i'})$
13:     $\boldsymbol{\theta}_l \leftarrow \boldsymbol{\theta}_l - \beta \nabla_{\boldsymbol{\theta}_l} \sum_{\mathcal{T}_i} \mathcal{L}_{\mathcal{T}_i}(f_{\boldsymbol{\theta}_i'})$
14: **end while**

---

first generate two lookahead models by updating the current model parameters $\boldsymbol{\theta}^{t-1}$ using momentum $\boldsymbol{m}^{t-1}$ and new gradients $\boldsymbol{g}^t$ separately in line (6) and line (7), which results in $\boldsymbol{\theta}_m^{t-1}$ and $\boldsymbol{\theta}_g^t$. Then $f(\cdot)$ is applied to both models with $S_{\mathcal{T}}$, yielding loss changes for both lookahead models: $\Delta \mathcal{L}_m^{t-1}$ and $\Delta \mathcal{L}_g^t$. In line (10), the softmax operator is used to normalize two loss changes so that the aggregation coefficients sum to 1. The resulting coefficients are used to predict adaptive learning rates in line (11), and aggregate the momentum and new gradients in line (12). Finally, line (13) updates the model parameters using accumulated momentum $\boldsymbol{m}^t$.

With dynamic coefficients generated through lookahead, if either the momentum or new gradients lead to an ineffective optimization direction, the small coefficient in line (12) can mitigate the negative impact and place greater emphasis on more effective optimization directions. As a result, the double-lookahead strategy can stabilize the momentum update and accelerate the convergence speed.

## 3.4 Dynamic Class Weighting Scheme for Classification Tasks

In FSL classification tasks employing meta-learning, the meta-trained base model is significantly influenced by prior knowledge. As a result, certain classes with features closer to prior knowledge become easier to classify, leading to an imbalance that compromises the model's performance. To address this issue, we introduce a dynamic class weighting schema, which emphasizes classes contributing more to model optimization. We estimate a class's contribution by observing the changes in its loss. If a class has been well-learned, the loss will be stable, and minor loss changes will be observed. In contrast, if there is a substantial loss drop for a particular class, it may contribute more to future updates. Based on the loss changes for each class, we use a softmax operator to obtain aggregation weights for every class. The accumulated loss is the weighted summation of all class losses. This process is defined as follows:

$$\mathcal{L}_1^t, \ldots, \mathcal{L}_C^t = f_2(S_{\mathcal{T}}, \boldsymbol{\theta}^{t-1}), \tag{5}$$

$$w_1, \ldots, w_C = \sigma((\mathcal{L}_t^1, \ldots, \mathcal{L}_t^C)/T), \tag{6}$$

$$\mathcal{L}^t = \sum_{k=1}^{C} w_k \mathcal{L}_k^t, \tag{7}$$

where $f_2(\cdot)$ is a loss function that computes losses for each class based on support data $S_{\mathcal{T}}$, $C$ is the number of classes in task $\mathcal{T}$, and $T$ is the temperature used in the softmax operator $\sigma(\cdot)$. In the above schema, if the class $k$ has a larger loss $\mathcal{L}_k^t$, its corresponding aggregation weight $w_k$ will be larger, leading to a stronger optimization signal from class $k$. Thus, the class that needs to be more well-trained will obtain more attention, leading to more effective optimization.

Table 1: Comparison results using Convnet4 on Mini-ImageNet, TieredImageNet, and Cifar100 datasets. We report performances in terms of accuracy (%) with standard deviation.*Baseline MC uses a larger Convnet4 backbone network, and uses extra data regularization method.

| Dataset | Method | 5-way-1-shot | 5-way-5-shot |
|---|---|---|---|
| Mini-ImageNet | Meta-LSTM [30] | $43.44 \pm 0.77$ | $60.60 \pm 0.71$ |
| | MAML [7] | $48.70 \pm 1.75$ | $63.11 \pm 0.92$ |
| | Meta-SGD [18] | $50.47 \pm 1.87$ | $64.03 \pm 0.94$ |
| | MAML+ALFA [3] | $50.58 \pm 0.51$ | $69.12 \pm 0.47$ |
| | e3bm [20] | 53.2 | 65.1 |
| | Sparse-MAML [44] | $51.04 \pm 0.59$ | $68.05 \pm 0.84$ |
| | MAML+SiMT [42] | $51.49 \pm 0.18$ | $68.74 \pm 0.12$ |
| | MeTAL [40] | $52.63 \pm 0.37$ | $70.52 \pm 0.29$ |
| | Meta-AdaM (ours) | $\mathbf{52.00} \pm 0.49$ | $\mathbf{70.70} \pm 0.49$ |
| | MC* [25] | $55.73 \pm 0.94$ | $70.33 \pm 0.72$ |
| TieredImageNet | MAML [7] | $49.06 \pm 0.50$ | $67.48 \pm 0.47$ |
| | MAML+ALFA [3] | $53.16 \pm 0.51$ | $70.54 \pm 0.46$ |
| | e3bm [20] | 52.1 | 70.2 |
| | MAML+SiMT [42] | $52.51 \pm 0.21$ | $69.58 \pm 0.11$ |
| | MeTAL [40] | $54.34 \pm 0.31$ | $70.40 \pm 0.21$ |
| | Meta-AdaM (ours) | $\mathbf{53.93} \pm 0.49$ | $\mathbf{72.66} \pm 0.49$ |
| Cifar100 | e3bm [20] | 39.9 | 52.6 |
| | MAML [7] | $38.20 \pm 0.48$ | $49.94 \pm 0.49$ |
| | MAML+ALFA [3] | $39.77 \pm 0.48$ | $53.39 \pm 0.49$ |
| | Meta-AdaM (ours) | $\mathbf{41.11} \pm 0.49$ | $\mathbf{56.32} \pm 0.49$ |

## 3.5 Meta-AdaM

We present our proposed Meta-AdaM in Algorithm 2. Given a task distribution $p(\mathcal{T})$, our goal is to meta-learn an optimal model initialization $\boldsymbol{\theta}$ and an adaptive learning learner $\boldsymbol{\theta}_l$. In each iteration, we sample a batch of tasks from $p(\mathcal{T})$. For each sampled task, we fine-tune an adapted model $\boldsymbol{\theta}'$ and update $\boldsymbol{\theta}$ and $\boldsymbol{\theta}_l$ using Algorithm 1. During testing, we directly utilize $\boldsymbol{\theta}$ and $\boldsymbol{\theta}_l$ for a new task.

# 4 Experiments

We conduct experiments to evaluate the proposed Meta-AdaM using three benchmark datasets.

## 4.1 Experimental Settings

This section describes datasets and backbone models used to evaluate our Meta-AdaM.

**Datasets**. We evaluate the proposed methods using three datasets: Mini-ImageNet [10], Tiered-ImageNet [33], and Cifar100 [16] datasets. *Mini-ImageNet* dataset contains 100 classes with 600 samples per class. Each data sample is an $84 \times 84$ colored image. Following previous works [30], we split 100 classes into 64, 16, and 20 class groups for training, validation, and testing. The image size of *Tiered-ImageNet* is also $84 \times 84$. This dataset consists of 608 classes with 779,165 images. These 608 classes are further combined into 34 high-level classes. These high-level classes are divided into 20, 6, and 8 class groups for training, validation, and testing [31], respectively. Tiered-ImageNet considers the class similarity when divide the datatset to ensure that the distribution of train and test data is very different. *Cifar100* dataset for few-shot learning contains 60,000 images with 100 classes. The image size is $32 \times 32$. The 100 classes are divided into 60, 20, and 20 groups for training, validation, and testing. These datasets are bench-marking datasets in the FSL domain.

Table 2: Hyperparameters for experiment.

| Hyperpameter | Value |
|---|---|
| tasks batch size | 2 |
| inner learning rate $\eta$ | 0.01 |
| outer learning rate $\alpha$, $\beta$ | 0.001 |
| # inner fine-tune step | 5 |
| # training epochs | 100 |
| # outer steps in each epoch | 500 |

**Backbone models**. We evaluate the proposed Meta-AdaM using two backbone models: Convnet4 and Resnet12 [12]. *Convnet4* is a 4-layer Convolutional Neural Network with 32.9 thousand trainable

Table 3: Comparison results using ResNet12 on Mini-ImageNet, TieredImageNet, and Cifar100 datasets. We report accuracy (%) with standard deviation. *Baseline MC uses WRN-28-10 as backbone and extra data regularization method.

| Dataset | Method | 5-way-1-shot | 5-way-5-shot |
|---------|--------|--------------|--------------|
| Mini-ImageNet | MAML [7] | $58.37 \pm 0.49$ | $69.76 \pm 0.46$ |
| | MAML+ALFA [3] | $59.74 \pm 0.49$ | $77.96 \pm 0.47$ |
| | MAML+SiMT [42] | $56.28 \pm 0.63$ | $72.01 \pm 0.26$ |
| | Sparse-MAML [44] | $56.39 \pm 0.38$ | $73.01 \pm 0.24$ |
| | MeTAL[40] | $59.64 \pm 0.38$ | $76.20 \pm 0.19$ |
| | Meta-AdaM (ours) | $\mathbf{59.89} \pm 0.49$ | $\mathbf{77.92} \pm 0.43$ |
| | MC* [25] | $64.40 \pm 0.10$ | $80.21 \pm 0.10$ |
| Tiered-ImageNet | MAML [7] | $58.58 \pm 0.49$ | $71.24 \pm 0.43$ |
| | MAML+ALFA [3] | $64.62 \pm 0.49$ | $82.48 \pm 0.39$ |
| | MAML+SiMT [42] | $59.72 \pm 0.22$ | $74.40 \pm 0.90$ |
| | MeTAL[40] | $63.89 \pm 0.43$ | $80.14 \pm 0.40$ |
| | Meta-AdaM(ours) | $\mathbf{65.31} \pm 0.48$ | $\mathbf{85.24} \pm 0.35$ |
| | MC* [25] | $67.21 \pm 0.10$ | $82.61 \pm 0.10$ |
| Cifar100 | MAML [7] | $38.79 \pm 0.48$ | $51.65 \pm 0.49$ |
| | MAML+ALFA [3] | $40.88 \pm 0.48$ | $54.54 \pm 0.49$ |
| | Meta-AdaM (ours) | $\mathbf{41.12} \pm 0.49$ | $\mathbf{56.14} \pm 0.49$ |

parameters, while *Resnet12* is an 18-layer Convolutional Neural Network featuring 4 million trainable parameters. By employing these two models, we demonstrate that our proposed optimizer is effective on both small and large networks.

**Hyperparameters**. We show the hyperparamters in Table 2, which applies to each dataset and each backbone. To ensure a fair comparison, we follow the setting of [3] and report the test performance of the ensemble of top-5 performing models on the validation set.

## 4.2 Results Using A Small Network

We first evaluate our Meta-AdaM using the Convnet4 network on three datasets. The experiments are conducted under two settings: 5-way-1-shot and 5-way-5-shot, where $a$-way-$b$-shot means each task contains $a$ classes and there are $b$ training examples for each class. We compare our Meta-AdaM with Meta-LSTM [30], MAML [7], Meta-SGD [18], and MAML+ALFA [3] as main baseline. A summary of the comparison results can be found in Table 1. From the results, we can observe that our method achieves promising results on all three datasets under two settings. Under the 5-way-1-shot setting, our Meta-adaM outperforms MAML and MAML+ALFA by margins of 3.30% and 1.42% on Mini-ImageNet, 4.87% and 0.77% on TieredImageNet, and 2.91% and 1.34% on Cifar100, respectively. Under the 5-way-5-shot setting, our method outperforms MAML and MAML+ALFA by margins of 7.59% and 1.58% on Mini-ImageNet, 5.18% and 2.12% on TieredImageNet, and 6.38% and 2.93% on Cifar100, respectively.

These promising results demonstrate that our Meta-AdaM can consistently yield more effective optimization results on various benchmark datasets. Notably, the performance gains are even larger in the 5-way-5-shot setting. Given that both the momentum and the new gradients estimated from a few training examples could be noisy with only one training example per class, it is difficult to produce effective gradient updates from them. The power of Meta-AdaM is better reflected by slightly increasing the number of training examples. We also compare our methods with other recent research works, including e3bm [20], Sparse-MAML [44], MAML+SiMT [42] and MeTAL [40]. We can observe that our methods outperform the other methods in most of the settings.

## 4.3 Results Using A Large Network

In the previous section, we evaluated our Meta-AdaM using a small network, Convnet4, as the backbone model. However, optimizing a large model is more challenging due to the risk of overfitting. In this section, we conduct experiments on a larger backbone model, Resnet12, to further demonstrate the effectiveness of our Meta-AdaM. The evaluation is conducted under 5-way-1-shot and 5-way-5-

Table 4: Abalation study results using the Mini-ImageNet dataset on the Convnet4 backbone network.

| Momentum | Look-ahead | Adaptive lr | 5-way-1-shot | 5-way-5-shot |
|:---:|:---:|:---:|:---:|:---:|
| ✓ | ✓ | ✓ | 51.64±0.49 | 68.80±0.46 |
| ✓ | ✓ | ✗ | 49.30±0.49 | 64.09±0.47 |
| ✗ | ✗ | ✓ | 49.02±0.50 | 67.98±0.46 |
| ✓ | ✗ | ✗ | 48.28±0.49 | 63.27±0.48 |
| ✗ | ✗ | ✗ | 48.70±1.75 | 63.11±0.92 |

Table 5: Comparison result LSTM and MLP using Convnet4 on Mini-ImageNet, TieredImageNet, and Cifar100 datasets. We report performances in terms of accuracy (%) with standard deviation.

| Dataset | Method | 5-way-1-shot | 5-way-5-shot |
|:---:|:---:|:---:|:---:|
| Mini-ImageNet | Meta-Adam with MLP | **52.27**± 0.49 | 70.48 ± 0.49 |
| | Meta-AdaM with LSTM | 52.00 ± 0.49 | **70.70** ± 0.49 |
| TieredImageNet | Meta-Adam with MLP | 53.48 ± 0.49 | 72.60 ± 0.49 |
| | Meta-AdaM with LSTM | **53.93** ± 0.49 | **72.66** ± 0.49 |
| Cifar100 | Meta-Adam with MLP | 40.28 ± 0.49 | 54.96 ± 0.49 |
| | Meta-AdaM with LSTM | **41.11** ± 0.49 | **56.32** ± 0.49 |

shot settings using the Mini-ImageNet, Tiered-ImageNet, and Cifar100 datasets. The comparison results with MAML and MAML+ALFA are summarized in Table 3.

Under the 5-way-1-shot setting, we observe margins of 0.15%, 0.69%, and 0.24% improvements compared to previous state-of-the-art models on the Mini-ImageNet, TieredImageNet, and Cifar100 datasets, respectively. Under the 5-way-5-shot setting, our Meta-AdaM outperforms ALFA by margins of 2.76%, and 1.60% on TieredImageNet, and Cifar100 datasets respectively. On Mini-ImageNet, our method gives a equal performance compared to MAML+ALFA. Our method also outperforms other methods, including MAML+SiMT [42] and MeTAL [40]. The consistently promising results on the three benchmark datasets demonstrate that our Meta-AdaM is effective when used to optimize a larger backbone model.

## 4.4 Ablation Study

In this study, we introduce three novel components in our Meta-AdaM: momentum, double lookahead and an adaptive learning rate via a meta-learned learner. Also, we introduce a dynamic class-weighting scheme for loss functions specially designed for classification tasks. In this section, we evaluate the individual contributions of each component. To accomplish this, we conduct experiments in both 5-way-1-shot and 5-way-5-shot settings, utilizing Convnet4 as the backbone model on the Mini-ImageNet dataset.

### 4.4.1 Momentum, Double lookahead and Adaptive learning rate

Specifically, we test varying combinations of the first three components and assess the performance of the resulting optimizers. These comparative results can be summarized in Table 4. The results in Table 4 demonstrate that each component contributes to the overall performance. Specifically, heuristic aggregation of momentum does not significantly enhance the performance, with a 0.42% decrease observed in the 1-shot and a modest 0.16% increase in the 5-shot setting compared to the original MAML. However, incorporating momentum with a double look-ahead strategy leads to superior results over those from the momentum-only setting. This enhanced optimizer surpasses the original MAML by 0.6% in the 1-shot and 0.98% in the 5-shot settings.The adaptive learning rate also plays a crucial role. By solely applying this adaptive learning rate in the inner loop, the resultant optimizer outperforms the original MAML by 0.32% in the 1-shot and 4.87% in the 5-shot settings. When these three components are combined, the performance improvement is even more significant. The momentum with look-ahead results in a more stable and efficient training optimization process, thereby aiding the meta-learner in better estimating the adaptive learning rates. From these results, we see a performance boost of 2.94% in the 1-shot and 5.69% in the 5-shot settings. To further evaluate the effect of weight update history, we conduct additional experiments that substitute the LSTM network with a MLP network with same inputs. The results are summarized in Table 5. From

Table 6: Ablation Study for class dynamic weight using Convnet4 on Mini-ImageNet, TieredImageNet, and Cifar100 datasets

| Dataset | Method | 5-way-1-shot | 5-way-5-shot |
|---|---|---|---|
| Mini-ImageNet | ALFA | $50.58 \pm 0.51$ | $69.12 \pm 0.47$ |
| | ALFA+DW | $50.65 \pm 0.49$ | $70.02 \pm 0.45$ |
| | Meta-AdaM w/o DW | $51.64 \pm 0.49$ | $68.80 \pm 0.46$ |
| | Meta-AdaM (ours) | $\mathbf{52.00} \pm 0.49$ | $\mathbf{70.70} \pm 0.49$ |
| TieredImageNet | ALFA | $53.16 \pm 0.51$ | $70.54 \pm 0.46$ |
| | ALFA+DW | $53.54 \pm 0.49$ | $72.36 \pm 0.19$ |
| | Meta-AdaM w/o DW | $53.62 \pm 0.50$ | $71.57 \pm 0.49$ |
| | Meta-AdaM (ours) | $\mathbf{53.93} \pm 0.49$ | $\mathbf{72.66} \pm 0.49$ |
| Cifar100 | ALFA | $39.77 \pm 0.48$ | $53.39 \pm 0.49$ |
| | ALFA+DW | $40.79 \pm 0.19$ | $54.34 \pm 0.48$ |
| | Meta-AdaM w/o DW | $40.14 \pm 0.49$ | $54.36 \pm 0.50$ |
| | Meta-AdaM (ours) | $\mathbf{41.11} \pm 0.49$ | $\mathbf{56.32} \pm 0.49$ |

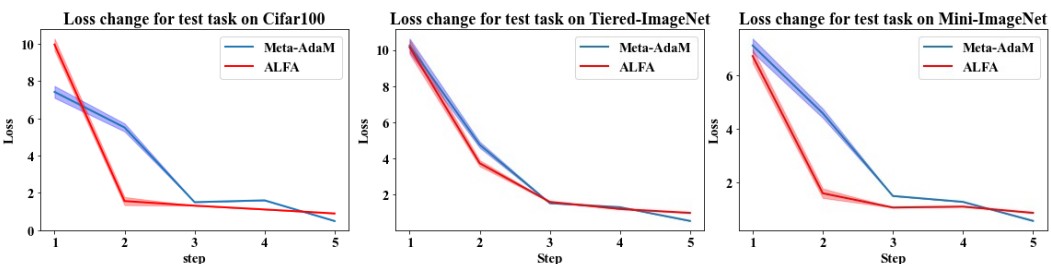

Figure 2: Visualization of loss changes for each step. The left, middle, and right figures are the average loss changes and 95% confidence interval for each step on Mini-ImageNet, Tiered-ImageNet, and Cifar100, respectively.

the result, LSTM outperform MLP network in 5 out of 6 experiment setting, which indicates that the weight update history is important.

### 4.4.2 dynamic class-weighting scheme

To demonstrate the effect of dynamic class, we conduct additional experiments on four configurations: ALFA, ALFA with Dynamic Class Weighting, Meta-AdaM excluding class weighting, and full Meta-AdaM and summarize in result in Table 6. The result shows that Meta-AdaM without Dynamic Class Weighting outperforms ALFA in 5 out of 6 evaluated settings. Noticeably, Dynamic Class Weighting provides a tangible enhancement when integrated with ALFA.

### 4.5 Visualization of Optimization Process

In this section, we visualize the optimization process of our Meta-AdaM and compare it with ALFA. We randomly select 900 testing tasks from three datasets: Mini-ImageNet, Tiered-ImageNet, and Cifar100. The experiment setting is 5-way-1-shot using Convnet4 as the backbone model. We use the checkpoints with the highest validation accuracy for Meta-AdaM and ALFA. We calculate the average loss change for all 900 tasks in each update step and plot the results in Figure 2. From the plot, our Meta-AdaM helps the model find a better optimum than ALFA.

Additionally, We conducted experiments on 300 5-way-1-shot test tasks on Cifar100 with Convnet4 as the backbone model. We include the change of inner learning rate $\eta$ and $\alpha$ and $\beta$ in the "Double look ahead strategy." in Figure 3. As illustrated in Figure 3 on the left, our optimizer tends to initially predict smaller learning rates due to the substantial conflict between momentum and gradient. A smaller learning rate will lead to a slow convergence rate in the first few steps, providing spaces for accumulating high-quality momentum. Therefore, from Figure 2, the loss from our method decreases slower than ALFA. Once the momentum quality is good after several steps, the predicted learning

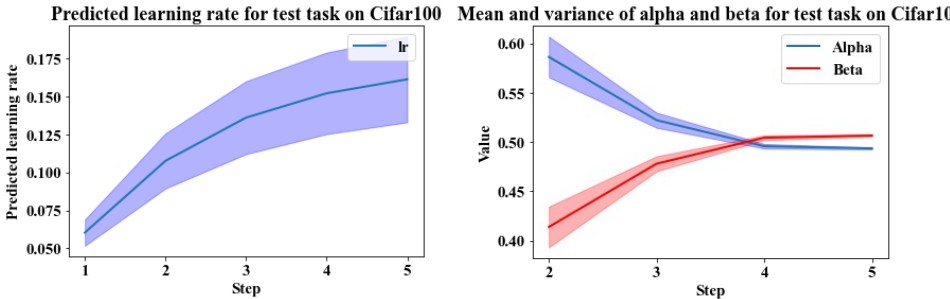

Figure 3: Left: Visualization of predicted learning rates in 5 steps with variance bonds. The predicted learning rates tend to increase along with the optimization process. Right: Visualization of computed coefficients $\alpha$ and $\beta$ of 5 steps with variance bonds in the fine-tuning process. The coefficients for momentum are small at the beginning due to their high variances.

rates become larger, leading to a better optimum than ALFA. This also highlights our contribution to using double-look-ahead to handle highly variable momentum.

Figure 3 on the right shows that the gradients give lower losses until the first three steps, so the gradient $\alpha$ coefficient is larger than for momentum $\beta$. However, after accumulating for a few steps, the accumulated momentum can generate a lower loss and a larger coefficient. Such results support the intuition that the momentum quality improves over the trajectory.

### 4.6 Efficiency Study of Meta-AdaM

In this section, we assess the efficiency of our Meta-AdaM. In Meta-AdaM, we employ a double lookahead strategy to evaluate the momentum and new gradients, which may introduce additional computational costs. To evaluate these added costs, we compare the total training time of MAML, MAML+ALFA, and Meta-AdaM. We use the experimental setting of 5-way-1-shot on the Mini-ImageNet dataset and record

Table 7: Average running time of one epoch.

| Method | Running time (s) |
|---|---|
| MAML | 252.04 |
| MAML+ALFA | 560.63 |
| Meta-AdaM | 595.91 |

the running time on an NVIDIA RTX A4000 GPU. The results are summarized in Table 7. The results indicate that our Meta-AdaM takes slightly longer running time than ALFA. However, considering the performance improvements brought by Meta-AdaM, the additional computational resources required by Meta-AdaM are justifiable.

### 4.7 Predicted Negative Learning Rate

During our experiments, the predicted learning rate in our adaptive learning rate component could occasionally be negative. This phenomenon has been described in [4] and observed in ALFA [3], which employs a two-layer network to predict learning rates. Interestingly, applying negative learning rates to trainable parameters in some update steps can lead to better optimization results. However, determining which parameters are suitable for a negative learning rate and when to apply the negative learning rate remains to be clarified. Further research exploring the role of negative learning rates in meta-learning may contribute to improving the optimization of meta-learning models.

## 5 Conclusion

In this work, we propose a meta-learning-specific optimizer, Meta-AdaM, which incorporates three novel optimization components that can better utilize update signals: momentum and new gradients. Specifically, we propose to meta-learn an adaptive learning rate learner using an LSTM network by looking back at the weight update history. We also introduce a double lookahead algorithm to dynamically accumulate momentum, leading to a more stable and effective momentum update and gradient update. Moreover, we propose a dynamic class weighting scheme to balance the losses across different classes. Experimental results demonstrate the effectiveness of our proposed Meta-AdaM.

# 6 Acknowledgement

This work has been supported in part by National Science Foundation grant III-2104797.

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

Table 8: Comparison results on few-shot regression tasks. The target function for regression is a sine curve y(x) = A sin(ωx). We report MSE error with a 95% confidence interval.

| Method | Training | 5-shot Testing | 10-shot Testing | 20-shot Testing |
|---|---|---|---|---|
| MAML[7] | 5-shot | $1.13 \pm 0.08$ | $0.85 \pm 0.14$ | $0.71 \pm 0.12$ |
| Meta-SGD[18] | 5-shot | $0.90 \pm 0.16$ | $0.63 \pm 0.12$ | $0.50 \pm 0.10$ |
| ALFA[3] | 5-shot | $0.60 \pm 0.04$ | $0.41 \pm 0.02$ | $0.25 \pm 0.22$ |
| Meta-AdaM | 5-shot | $\mathbf{0.52} \pm 0.04$ | $\mathbf{0.35} \pm 0.04$ | $\mathbf{0.23} \pm 0.01$ |
| MAML[7] | 10-shot | $1.17 \pm 0.16$ | $0.77 \pm 0.11$ | $0.56 \pm 0.08$ |
| Meta-SGD[18] | 10-shot | $0.88 \pm 0.14$ | $0.53 \pm 0.09$ | $0.35 \pm 0.06$ |
| ALFA[3] | 10-shot | $0.72 \pm 0.16$ | $0.45 \pm 0.03$ | $\mathbf{0.26} \pm 0.02$ |
| Meta-AdaM | 10-shot | $\mathbf{0.66} \pm 0.22$ | $\mathbf{0.34} \pm 0.05$ | $\mathbf{0.23} \pm 0.01$ |
| MAML[7] | 20-shot | $1.29 \pm 0.20$ | $0.76 \pm 0.12$ | $0.48 \pm 0.08$ |
| Meta-SGD[18] | 20-shot | $1.01 \pm 0.17$ | $0.54 \pm 0.08$ | $0.31 \pm 0.05$ |
| ALFA[3] | 20-shot | $1.01 \pm 0.18$ | $0.48 \pm 0.09$ | $0.25 \pm 0.03$ |
| Meta-AdaM | 20-shot | $\mathbf{0.95} \pm 0.11$ | $\mathbf{0.43} \pm 0.06$ | $\mathbf{0.23} \pm 0.03$ |

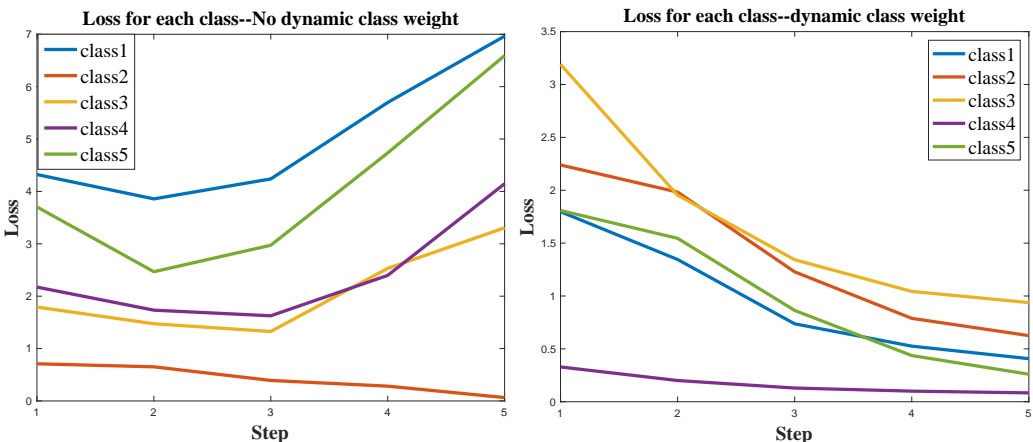

Figure 4: Visualization of dynamic class weighting scheme. The left and right figures show the optimization process without and with the dynamic class weighting scheme, respectively.

# 7 Appendix

## 7.1 Result on regression task

To further prove the efficiency of our method, we conduct additional experiments on few-shot regression tasks and show the result in Table 8. We follow the experiment setting of the Meta-SGD[18]. The regression task is to map the underlying function based on the input. The target function is $y(x) = A \sin(\omega x)$, where amplitude $A$, frequency $\omega$, and phase $b$ all follow a uniform distribution with the interval [0.1, 5.0], [0.8, 1.2], and [0, $\pi$]. We train three few-shot regression learners on 5-shot, 10-shot, and 20-shot tasks separately. Each learner will test on 100 5-shot, 10-shot, and 20-shot testing tasks. We report MSE errors with a 95% confidence interval.

We compare our approach with MAML[7], Meta-SGD[18], Meta-SGD+ALFA[3], and Meta-SGD + META-AadM. The result has shown that our method outperforms the baselines, Meta-SGD and Meta-SGD+ALFA, in all settings. The promising results indicate that our proposed method is also effective on few-shot regression tasks.

## 7.2 Study of the Dynamic Class Weighting Scheme

In previous section, we present quantitative results to demonstrate the effectiveness of the dynamic class weighting scheme. In this section, we visualize the loss changes for each class when using this scheme. We compare the loss function using our scheme and the one without a class weighting scheme. Each optimization process consists of five update steps. The left figure in Figure 4 displays

the loss changes of each class without employing a class weighting scheme. Class 2 has a significantly lower loss than the other classes. For the last four steps, only the loss of class 2 is optimized, while the other four classes remain under-optimized. Such situations can lead to model collapse problems in meta-learning [26], which means the model tends to classify all testing examples into one or two classes. The right figure in Figure 4 illustrates the loss changes when using the dynamic class weighting scheme. Class 4 has a lower loss in the first step compared to the other classes. The optimizer focuses more on the other four classes by emphasizing those with larger losses. Ultimately, all class losses converge towards the optimization endpoint, effectively mitigating the risk of model collapse problems.

