# OpenReview forum: "Meta-AdaM: An Meta-Learned Adaptive Optimizer with Momentum for Few-Shot Learning"
_NeurIPS.cc/2023/Conference — NeurIPS 2023 poster_

### Official Review · Reviewer_n9ad · 2023-06-23

**Soundness:** 3 good
**Presentation:** 2 fair
**Contribution:** 3 good
**Rating:** 6
**Confidence:** 4

**Summary:**

The paper presents Meta-AdaM, a meta-learned adaptive optimizer that also includes momentum. The parameter of the optimizer in the inner loop, i.e. inner learning rate and momentum, are predicted by a LSTM network, that takes into account the previous gradient and momentum steps. The optimization process also includes a double look-ahead strategy for the prediction of the coefficients for the gradient and the momentum. Lastly, the optimization also includes a dynamic class weighting scheme, using a softmax to assign weights to each class according to the loss changes for each class.
The optimizer is then combined with MAML and evaluated on three benchmarks datasets (mini-Imagenet, tiered-Imagenet and CIFAR100) with two different backbones (conv4 and Resnet12).

**Strengths:**

### Originality
The paper builds upon ALFA [1] by replacing the learned network that predicts the *inner learning rate* and *weight decay* parameters, by a LSTM network, that predicts *inner learning rate* and *momentum* parameters. The LSTM allows taking into account previous update steps. The dynamic class weighting scheme is also an interesting regularization approach to reduce overfitting to easy classes in the inner loop.
### Significance
The proposed method shows strong results on the different benchmarks considered.

[1] : Baik, S., Choi, M., Choi, J., Kim, H., & Lee, K. M. (2023). Learning to learn task-adaptive hyperparameters for few-shot learning. IEEE Transactions on Pattern Analysis and Machine Intelligence.

**Weaknesses:**

### Clarity

- The paper is difficult to follow as there is a lot of text to describe the method, but it is missing a formal definition of the process. Thankfully, the two algorithms help to clarify the text, but the equations in the algorithms could be directly introduced and described in the text. There is also several concepts vaguely defined such as the loss function $f_2$ in Equation 5 and the temperature $T$ in Equation 6 that is never mentioned in the hyperparameters.
- The discussion around the dynamic class weighting scheme seems unrelated to the original idea described throughout the paper. The way it is introduced in the optimization is also not really explained, neither in the text nor in the algorithms.
- The ablation study mentions four different components, but it is not clear what the *momentum* component refers to.
### Quality
- The gradient descent formula described in Equation 1 and 2 is not the commonly used one, described in [2] for instance. Here, the learning rate is impacting the update coming from the momentum.
- In line 175, it is said that the coefficients $\alpha$ and $\beta$ should sum to 1, hence the use of a softmax function, but it is not clear to me why it should be the case, and it is never explained.
### Significance
The introduction of the dynamic class weighting scheme, while being an interesting addition, makes the work specific to the problem of few-shot classification, even though this component does not seem necessary in the approach. From Table 4 in the ablation study, we can see that it has a strong impact on the performance, mainly in the 5-shot setting, and makes the method more effective than others in the mini-imagenet benchmark. Results without this weighting and in other tasks, such as few-shot regression, would make the paper and the evaluation of the approach more solid.

[1] : Baik, S., Choi, M., Choi, J., Kim, H., & Lee, K. M. (2023). Learning to learn task-adaptive hyperparameters for few-shot learning. IEEE Transactions on Pattern Analysis and Machine Intelligence.

[2] : Sutskever, I., Martens, J., Dahl, G., & Hinton, G. (2013, May). On the importance of initialization and momentum in deep learning. In International conference on machine learning (pp. 1139-1147). PMLR.

**Questions:**

- From the paper and the equations, it seems that the optimization process does not include weight decay. This difference from the ALFA [1] paper is not really explained in the related work section, what is the motivation behind this change ?
- Why $\alpha$ and $\beta$ need to sum to 1 ? Why the learning rate is included in the update coming from the momentum ?
- How the dynamic class weighting scheme is included in the optimization process ? Does it impact both the loss of the LSTM and the base learner ?
- What are the results without the class weighting ? And in different tasks than classification ?

**Limitations:**

The authors did not include any discussion of the limitations of their work, nor of potential negative impacts. An important limitation is the limitation to few-shot classification tasks as mentioned above.

---

> ### Author Rebuttal · Authors · 2023-08-09
>
> **Table 1:** Ablation Study for class dynamic weight using Convnet4 on Mini-ImageNet, TieredIma- geNet, and Cifar100 datasets
> | Dataset|Method|5-way-1-shot| 5-way-5-shot |
> | :----: | :-----:|:--------------:|:-------------:|
> |Mini-ImageNet|ALFA |50.58 ± 0.51|69.12 ± 0.47|
> |Mini-ImageNet|ALFA+DW| 50.65 ± 0.49|70.02 ± 0.45|
> |Mini-ImageNet|Meta-AdaM w/o DW| 51.64 ± 0.49| 68.80 ± 0.46|
> |Mini-ImageNet|Meta-AdaM| 52.00 ± 0.49| 70.70 ± 0.49|
>
> | Dataset|Method|5-way-1-shot| 5-way-5-shot |
> | :----: | :-----:|:--------------:|:-------------:|
> |TieredImageNet|ALFA |53.16 ± 0.51| 70.54 ± 0.46|
> |TieredImageNet|ALFA+DW| 53.54 ± 0.9|72.36± 0.19|
> |TieredImageNet|Meta-AdaM w/o DW| 53.62 ± 0.50|71.57 ± 0.49|
> |TieredImageNet|Meta-AdaM| 53.93 ± 0.49|72.66 ± 0.49|
>
> | Dataset|Method|5-way-1-shot| 5-way-5-shot |
> | :----: | :-----:|:--------------:|:-------------:|
> |Cifar100|ALFA|39.77 ± 0.48|53.39 ± 0.49|
> |Cifar100|ALFA+DW|40.79 ± 0.19|54.34 ± 0.48|
> |Cifar100|Meta-AdaM w/o DW|40.14 ± 0.49|54.36 ± 0.50|
> |Cifar100|Meta-AdaM (ours)|41.11 ± 0.49|56.32 ± 0.49|
> ____
> **Concern 1 (Clarity 1):**
>
> *The paper is difficult to follow as there is a lot of text to describe the method, but it is missing a formal definition of the process.*
>
> **Answer:**
>
> Thank you for the suggestions. We will improve the clarity of our work and provide comprehensive formal definitions for variables used in the work.
>
> ___
> **Concern 2 (Clarity 2):**
>
> *The discussion around the dynamic class weighting scheme seems unrelated to the original idea described throughout the paper. The way it is introduced in the optimization is also not really explained, neither in the text nor in the algorithms.*
>
> **Answer:**
>
> The dynamic class weighting aims to provide a more balanced training signal. The loss computed by it is used to train both LSTM and the base learner.
>
> ___
>
> **Concern 3 (Clarity 3):**
>
> *The ablation study mentions four different components, but it is not clear what themomentum component refers to.*
>
> **Answer:**
>
> Sorry for the confusion. The momentum component indicates whether we keep an accumulated momentum for the inner loop for weights update and adaptive learning rate prediction.
>
> ___
> **Concern 4 (Quality 1):**
>
> *The gradient descent formula described in Equation 1 and 2 is not the commonly used one, described in [2] for instance. Here, the learning rate is impacting the update coming from the  momentum.*
>
> **Answer:**
>
> Thanks for pointing out this. We want to note that Equation 1 represents the gradient descent. We substitute the gradient $g_{i}^{t}$ in SGD with accumulated first-order momentum. Equation 2 represents the update $\alpha_i^t \times g_{i}^{t} + \beta_i^t \times m_{i}^{t-1}$ process of accumulated first-order momentum. This form is widely used in popular optimizers for deep learning models, such as AdaM [3], AdaMax [3], and RmsDrop [2].
> ___
>
> **Concern 5 (Question 2):**
>
> *Why α and β need to sum to 1? Why the learning rate is included in the update coming24
> from the momentum?*
>
> **Answer:**
>
>  The primary rationale behind setting the sum of α and β to 1 is to ensure that the resultant gradient aligns in scale with both the momentum and the original gradient. This approach follows the same strategies employed by popular optimizers frequently used by deep learning models, such as AdaM [3], AdaMax [3], and RmsDrop [2]. Incorporating the learning rate with the momentum arises because momentum isn’t directly applied to weight updates. As observed in algorithms like Adam, the learning rate and momentum
> are combined.
> ____
>
> **Concern 6 (Significance & Question 4):**
>
> *Contribution of dynamic class weighting scheme*
>
> **Answer:**
>
> Thank you for your insightful recommendation. We conduct experiments and provide more comparative results with ALFA [1] in Table 1. Our comparison spans four configurations: ALFA, ALFA with Dynamic Class Weighting, Meta-AdaM excluding class weighting, and the full Meta-AdaM. Based on the results, it’s shown that our Meta-AdaM without dynamic weighting surpasses ALFA in 5 out of the 6 evaluated settings. Moreover, Dynamic Class Weighting offers a tangible enhancement when integrated with ALFA.
>
> For few-shot regression tasks, the dynamic class weight scheme cannot be used since it is used to balance the performances of different classes. While in regression tasks, there is no explicit class information.
>
> ___
>
> **Question 1:**
>
> *From the paper and the equations, it seems that the optimization process does not include weight decay. This difference from the ALFA [1] paper is not really explained in the related work section, what is the motivation behind this change?*
>
> **Answer:**
>
>  Thank you for drawing attention to this distinction. Notably, our method can predict weight decay, as illustrated in Eq. (3). This potential enhancement could positively influence the inner optimization process owing to its inherent regularization effects. In the current context, we emphasize the influences of adaptive learning rates and momentum. By sidelining the impact of weight decay, we can observe their contributions more clearly. Nonetheless, we are considering the inclusion of weight decay prediction from Eq. (3) in our finalized version.
> ____
>
> **Question  3:**
>
> *How the dynamic class weighting scheme is included in the optimization process? Does it impact both the loss of the LSTM and the base learner?*
>
> **Answer:**
>
>  Thank you for highlighting that. Indeed, you’re right. The LSTM and the base learner will be trained using the dynamic loss.
> ___
>
> **References:**
>
> [1] S. Baik, M. Choi, J. Choi, H. Kim, and K. M. Lee. Meta-learning with adaptive hyperparameters,2020.
>
> [2] I. Goodfellow, Y. Bengio, and A. Courville. Deep learning. MIT press, 2016.
>
> [3] D. P. Kingma and J. Ba. Adam: A method for stochastic optimization, 2017.

---

> > ### Comment · Reviewer_n9ad · 2023-08-14
> >
> > Thank you for the detailed answer and the additional results.
> > - I had a misunderstanding on the gradient descent formula, and the answer provided helped me clarify it. From the answer, I understand that the authors consider an *adaptive learning rate strategy with momentum*, such as the Adam or RMSProp optimizers. It also explains why $\alpha$ and $\beta$ sums to 1.
> > - I appreciate the results without the dynamic class weighting (DCW) and the comparison with ALFA + DCW. We can see now that the method has strong performance even without it. I highly encourage the authors to include these results in the revised version.
> >
> > **Remaining concerns**
> > - I still think that additional results on regression tasks would strengthen the contribution. I understand though that running these experiments might be difficult given the time. I'm fully aware that DCW cannot be used in this setting, but I don't think that DCW should be seen as an *intrinsic component* of the proposed method. As shown in the provided table, the method has good results even without DCW, so it should also translate to the regression case, similarly to ALFA.
> > - I'm not sure to understand how weight decay can be included from Eq. (3). I think that it could be an interesting addition and discussion to the contribution.
> >
> > Given that my misunderstanding has been clarified, I'm increasing my rating to "borderline accept". I would be happy to increase again my rating if the authors can provide results on regression tasks.

---

> > > ### Author Response · Authors · 2023-08-16
> > > **Few-shot regression results**
> > >
> > > **Table 1:** *Comparison results on few-shot regression tasks. The target function for regression is a sine
> > > curve $y(x) = A \sin(\omega x)$. We report MSE errors with a 95% confidence interval.*
> > > |Method |Training |5-shot Testing| 10-shot Testing| 20-shot Testing|
> > > |:-------|:---------|:---------------|:----------------|:------------------|
> > > |MAML| 5-shot |1.13 ± 0.08|0.85 ± 0.14| 0.71 ± 0.12|
> > > |Meta-SGD| 5-shot |0.90 ± 0.16 |0.63 ± 0.12| 0.50 ± 0.10|
> > > |Meta-SGD+ALFA (reproduced)| 5-shot| 0.60 ± 0.04| 0.41 ± 0.02| 0.25 ± 0.22|
> > > |Meta-SGD+Meta-AdaM| 5-shot| **0.52** ± 0.04 | **0.35** ± 0.04|  **0.23** ± 0.01|
> > >
> > > |Method |Training |5-shot Testing| 10-shot Testing| 20-shot Testing|
> > > |:-------|:---------|:---------------|:----------------|:------------------|
> > > |MAML| 10-shot| 1.17 ± 0.16| 0.77 ± 0.11| 0.56 ± 0.08|
> > > |Meta-SGD| 10-shot| 0.88 ± 0.14| 0.53 ± 0.09| 0.35 ± 0.06|
> > > |Meta-SGD+ALFA (reproduced)| 10-shot| 0.72 ± 0.16| 0.45± 0.03| 0.26 ± 0.02|
> > > |Meta-SGD+Meta-AdaM| 10-shot| **0.66** ± 0.22| **0.34** ± 0.05| **0.23** ± 0.01|
> > >
> > >
> > > |Method |Training |5-shot Testing| 10-shot Testing| 20-shot Testing|
> > > |:-------|:---------|:---------------|:----------------|:------------------|
> > > |MAML| 20-shot |1.29 ± 0.20| 0.76 ± 0.12| 0.48 ± 0.08|
> > > |Meta-SGD| 20-shot| 1.01 ± 0.17| 0.54 ± 0.08| 0.31 ± 0.05|
> > > |Meta-SGD+ALFA (reproduced)| 20-shot| 1.01± 0.18 |0.48± 0.09| 0.25 ± 0.03|
> > > |Meta-SGD+Meta-AdaM| 20-shot| **0.95** ± 0.11| **0.43** ± 0.06| **0.23** ± 0.03|
> > > ___
> > >
> > > **Concern 1**:
> > >
> > > *Provide results on regression tasks.*
> > >
> > > ___
> > > **Answer:**
> > >
> > > Thanks for your quick response. To address your concern about few-shot regression tasks, we conduct additional experiments on few-shot regression tasks and show the results in Table 1. We follow the experimental settings of Meta-SGD [2]. The regression tasks are to map the underlying functions based on the input. The target function is $y(x) = A sin(\omega x)$, where amplitude $A$, frequency $\omega$, and phase $b$ follow a uniform distribution with intervals [0.1, 5.0], [0.8, 1.2], and [0, $\pi$]. We train three few-shot regression learners on 5-shot, 10-shot, and 20-shot tasks separately. Each learner will test on 100 5-shot, 10-shot, and 20-shot testing tasks. We report MSE errors with a 95% confidence interval.
> > >
> > > We compare MAML, Meta-SGD, Meta-SGD+ALFA[1], and Meta-SGD+Meta-AdaM.
> > > The results show that our method outperforms two major baselines, Meta-SGD and Meta-SGD+ALFA
> > > in all settings. The promising results indicate that our proposed method is also effective on few-shot regression
> > > tasks.
> > > ___
> > > **Reference**:
> > >
> > > [1] S. Baik, M. Choi, J. Choi, H. Kim, and K. M. Lee. Meta-learning with adaptive hyperparameters,2020.
> > >
> > > [2] Z. Li, F. Zhou, F. Chen, and H. Li. Meta-sgd: Learning to learn quickly for few-shot learning,2017.2

---

> > > > ### Comment · Reviewer_n9ad · 2023-08-21
> > > >
> > > > Thank you again for the detailed results on few-shot regression tasks, I encourage the authors to include them in the final version of the paper.
> > > > As all my concerns have been addressed, I'm increasing my rating to "weak accept".

---

### Official Review · Reviewer_7eNt · 2023-07-05

**Soundness:** 2 fair
**Presentation:** 4 excellent
**Contribution:** 3 good
**Rating:** 6
**Confidence:** 5

**Summary:**

The authors introduce an adaptive optimizer for meta-learning that uses double lookahead to incorporate momentum in scenarios where ‘conventional’ approaches of using momentum usually fall short (few-shot inner-loop, due to high momentum/gradient fluctuations during initial steps). The use of an LSTM to predict the adapted learning rates allows for consideration of the weight-update history.  To further boost the obtained results, the authors additionally introduce dynamic weighting of the class-specific losses to aid a more balanced optimization / loss minimization.

**Strengths:**

* Well written paper, easy to read and follow, good introduction of method and required background. Placement within related work well accomplished, and approach well motivated.
* Generally interesting idea and approach: Incorporating momentum with the double look-ahead and predicting the learning rate based on the history are to me the main contributions.
* Results of the overall approach seem solid.
* For the most part, interesting evaluations including run-time.


**Weaknesses:**

Major:

1. **Entanglement of ‘dynamic class weighting scheme’ and ‘optimiser approach’**:
 While an interesting contribution in itself, the “dynamic class weighting scheme” is highly entangled with the actual meta-adam approach throughout the reported results. While I do understand the pressure to ‘outperform’ other works, the reported results in its current state do in my opinion not allow for an insightful comparison of approaches & limits the reader’s insights in an unnecessary manner. I would highly encourage the authors to:
   - Add the Meta-AdaM (w/o dyn-cl-w) to the tables
   - Evaluate MAML or others together with their proposed dyn-cl-w.

   This would allow the reader to better judge the influence of the individual contributions and their influence beyond this particular work!

   Note: I am aware that Table 4 shows the result w/o dynamic class loss, but since the *paper’s main selling point and TITLE focuses on the optimiser*, this should be individually demonstrated and be part of an honest discussion.

2. **Usefulness of the LSTM / history**:
  How essential is the fact that the adaptive learning rate being predicted by an LSTM? How important is the history, really – which is one of the main claims of the paper! -> How does it compare to e.g. using an MLP with similar complexity and the same input (prev. momentum and current gradient)?


Minor:

3. **Dataset inconsistency** (Description & References):
  The authors state 'Cifar100' as few-shot learning benchmark – however, there two inconsistencies that should be improved:
   * The FSL community uses two different versions: Cifar-FS and FC100, that yield very different results due to their different complexities. The provided reference in the paper links neither of them, but the original non-FSL Cifar  -> clarification here is required.
   * The description of the dataset (sec 4.1) states “*with 100 different labels of birds*”, which seems more like a description for the popular CUB dataset than Cifar -- I assume some mixup has happened here?


**Questions:**

The three main points & questions are listed in the weaknesses section. However, I have several additional suggestions/questions I'd like comments from the authors on:

Several experiments / ablation studies would imo strengthen this work:
* What are the actual learning rates predicted within the inner loop? (Is there some ‘commonality’ within the same step across different test episodes?)
* What is the ‘average’ mix of gradient and momentum contribution across inner-loop update steps (i.e. Algorithm 1, l.10)? Does this reflect  ‘intuitions’ of growing momentum along the trajectory, or maybe defy these? (visualization/comment would be interesting)
* How big is the influence of the base learning rate in Algorithm 1? You are using it in lines 6-9 to predict the effect of taking a step of this size into the gradient or momentum direction – the step-size (and direction) of the actual step then taken however differs and is according to your comments ‘not limited’ (not even sign-wise) -> Is the lookahead still a ‘good and robust’ measure, or did you encounter difficulties? Does it change when choosing larger / smaller base-lrs, or is it rather robust (and why)?

Additional questions:
* What is the size of the LSTM, and how much complexity does it add?
* How is the temperature chosen for the dynamic class weighting, and how much does it matter & potentially differ across datasets?
* I’m slightly confused about the message of Figure 2: While Meta-AdaM does converge to a better minimum/lower loss at the final step, ALFA seems to converge faster – It would be helpful for the authors to comment on this.  (In its current state, I am not convinced about this part of the paper in general, since we already see in the tables that Meta-Adam seems to converge to a better minimum, and the only fact the figure shows me is that it does so in a slower way than ALFA?)
* Algorithm 2 ln 12/13 indicate SGD for outer loop – is this a simplification for conciseness, or did you choose SGD for the outer loop?


Some remarks to further improve presentation:
* Stating where the results for MAML are taken from would be helpful, since some of the datasets and architectures have not been evaluated in the original paper
* Line 36/37: “[…] has been shown that weight-update history is more important than the weights themselves […]” -> Reference would be important! I am aware that some are given in line 131, but two entire books without any sub-indication are a rather large search space for the reader! -> more recent publication or chapter/sub-chapter of book would be appreciated
* Sorting the references in ascending order (easier for reader to cross-check)
* Typos/Layout: Fig. 1: space after “A”; mix of capitalized and non-capital words within sub-captions inconsistent

**Note**: I do think the idea presented in this paper is interesting! Depending on the response from the authors, I am happy to reconsider my current rating.

> UPDATE: raised rating to weak accept after rebuttal.


**Limitations:**

The authors provide the runtime in comparison to related approaches, which adequately addresses the main limitation I see for this approach.

---

> ### Author Rebuttal · Authors · 2023-08-09
>
> **Table 1:** Ablation Study for class dynamic weight using Convnet4.
> | Dataset|Method|5-way-1-shot| 5-way-5-shot |
> | :----: | :-----:|:--------------:|:------------:|
> |Mini-ImageNet|ALFA |50.58 ± 0.51|69.12 ± 0.47|
> |Mini-ImageNet|ALFA+DW| 50.65 ± 0.49|70.02 ± 0.45|
> |Mini-ImageNet|Meta-AdaM w/o DW| 51.64 ± 0.49| 68.80 ± 0.46|
> |Mini-ImageNet|Meta-AdaM| 52.00 ± 0.49| 70.70 ± 0.49|
> |TieredImageNet|ALFA |53.16 ± 0.51| 70.54 ± 0.46|
> |TieredImageNet|ALFA+DW| 53.54 ± 0.9|72.36± 0.19|
> |TieredImageNet|Meta-AdaM w/o DW| 53.62 ± 0.50|71.57 ± 0.49|
> |TieredImageNet|Meta-AdaM| 53.93 ± 0.49|72.66 ± 0.49|
> |Cifar100|ALFA|39.77 ± 0.48|53.39 ± 0.49|
> |Cifar100|ALFA+DW|40.79 ± 0.19|54.34 ± 0.48|
> |Cifar100|Meta-AdaM w/o DW|40.14 ± 0.49|54.36 ± 0.50|
> |Cifar100|Meta-AdaM (ours)|41.11 ± 0.49|56.32 ± 0.49|
>
> **Table 2**: Comparison result LSTM and MLP using Convnet4.
> | Dataset|Method|5-way-1-shot| 5-way-5-shot |
> | :----: | :-----:|:--------------:|:------------:|
> |Mini-ImageNet|Meta-Adam with MLP| 52.27± 0.49| 70.48 ± 0.49|
> |Mini-ImageNet|Meta-AdaM with LSTM|52.00 ± 0.49| 70.70 ± 0.49|
> |TieredImageNet| Meta-Adam with MLP|53.48 ± 0.49|72.60 ± 0.49|
> |TieredImageNet|Meta-AdaM with LSTM|53.93 ± 0.49|72.66 ± 0.49|
> |Cifar100|Meta-Adam with MLP |40.28 ± 0.49 |54.96 ± 0.49|
> |Cifar100| Meta-AdaM with LSTM| 41.11 ± 0.49 |56.32 ± 0.49|
>
> **Weakness 1:**
> *Add the Meta-AdaM (w/o dyn-cl-w) to the tables. Evaluate MAML or others together with proposed dyn-cl-w.*
>
> **Answer**
> We conduct additional experiments on four configurations: ALFA [1], ALFA with Dynamic Class Weighting, Meta-AdaM excluding class weighting, and Meta-AdaM in Table 1. The results show that Meta-AdaM without Dynamic Class Weighting outperforms ALFA in 5 out of 6 evaluated settings. Notably, Dynamic Class Weighting can enhance other meta-learners like ALFA.
> ___
>
> **Weakness 2**
> *How essential the adaptive learning rate is predicted by LSTM? How important is history-> How does it compare to e.g. using an MLP with similar complexity and the same input (prev. momentum and current gradient)?*
>
> **Answer**
> We conduct additional experiments on LSTM and an MLP with the same inputs: accumulated momentum and gradients. The result is presented in Table 2. From the results, we can observe that LSTM outperforms MLP in most settings. This indicates the importance of weight update history.
> ___
> **Weakness 3**
> *Dataset inconsistency*
>
> **Answer**
> Thanks for pointing out that. We will change to the right description in our final paper.
> ___
>
> **Question 1:**
> *What are the actual learning rates predicted within the inner loop? Is there some commonality within the same step across different test episodes*
>
> **Answer**
>  We conduct experiments on test tasks on Cifar100 with Convnet4 as the backbone model and present the result in Figure 2 in the global rebuttal. From the result, we can observe the predicted learning rate is small at the beginning since the conflict between momentum and gradient is large in the first few steps. The learning rate becomes larger along with the update steps. This
> behavior indicates the conflict between momentum and gradient is smaller, and we get a smoother update direction.
> ___
>
> **Question 2**
> *What is the average mix of gradient and momentum across inner-loop update steps? Does this reflect intuitions of growing momentum*
>
> **Answer**
> We conduct experiments on test tasks on Cifar100 with Convnet4 as the backbone and present the results in Figure 2. The result shows that the gradients give lower losses until the first three steps. So the coefficient for gradient α is larger than for momentum β. However, after accumulating for a few steps, the accumulated momentum can generate a lower loss and a larger coefficient. Such results support the intuition that the momentum quality improves over the trajectory.
> ___
>
> **Question3**
> *How big is the influence of the base learning rate in Algorithm 1*
>
> **Answer**
> In our experiments, we set the base learning rate as 0.01. We do some additional evaluation on base learning rates 0.001 and 0.0001. Under the 5-way-1-shot setting on Mini-Imagenet using Convnet4 as the backbone, the accuracy for learning rates of 0.001 and 0.0001 are similar, which indicates that LSTM can adjust the predicted learning rate accordingly. The base learning rate has a small impact on the overall performance.
> ___
> **Question 4**
> *What is the size of LSTM, and how much complexity does it add*
>
> **Answer**
> The LSTM used in our paper contains two layers with 30 and 50 hidden sizes for 5-way-1-shot and 5-way-5-shot tasks, respectively. The complexity will increase. Table 5 in our paper shows that our method has an extra 35-second running time compared to ALFA for one epoch.
> ___
>
> **Question 5**
> *How is temperature chosen*
>
> **Answer**
> In our experiment, we choose 15 for 5-way-1-shot tasks and 25 for 5-way-5-shot tasks during the dynamic class weighting in all our settings and datasets.
> ___
>
> **Question 6**
> *Why ALFA seems to converge faster*
>
> **Answer**
> For the convergence rate of the first several steps, we empirically show the predicted learning rates across test tasks in Figure 2 in the global rebuttal. The results show that our optimizer tends to predict a smaller learning rate at the first few steps since the conflict between momentum and gradient is large. This can be reflected in Figure 3 in the global rebuttal,
> where momentum coefficients are smaller than gradients. A smaller learning rate will lead to a slow convergence rate in the first few steps, providing spaces for accumulating high-quality momentum. Once the momentum quality is good after several steps, the predicted learning rates become larger, leading to a better optimum than ALFA.
> ___
>
> **Question 7**
> *did you choose SGD for the outer loop?*
>
> **Answer**
> In our experiment, we use Adam as our outer loop optimizer. In Algorithm 2 ln 12/13, we use a general update form.
> ___
> **References**
>
> [1] S. Baik, M. Choi, J. Choi, H. Kim, and K. M. Lee. Meta-learning with adaptive hyperparameters,2020

---

> > ### Comment · Reviewer_7eNt · 2023-08-13
> >
> > I'd like to thank the authors for their thorough response!
> >
> > The provided answers & results have addressed most of my questions -- I appreciate the honesty regarding the provided results, especially regarding the clarification of the dynamic class reweighting scheme.
> >
> > I think the insights provided during the rebuttal are important and the authors should make a deliberate effort on incorporating them into the paper where possible (and add references to the supplementary material where space is too limited).
> >
> > However, I do keep my concern that while the presented method of momentum-incorporation and weight-update-history are interesting aspects, the dynamic class weighting has a major influence on the presented results -- and the paper in its current form more-or-less hides this fact (as pointed out by other reviewers as well);
> > Additionally, to claim to present an optimizer for "few shot learning" as stated in the title, I have to agree with other reviews that 'regression' should at least be demonstrated as well -> Note that ALFA [1] does present regression results (Sec. 4.5).
> >
> > All in all, I do see the novelty but equally the remaining limitations of the work in its current form.
> >
> > ->  I update my rating to 'weak accept'  (would support acceptance due to novelty; but not champion this paper)

---

> > > ### Author Response · Authors · 2023-08-16
> > > **Few-shot regression results**
> > >
> > > **Table 1:** *Comparison results on few-shot regression tasks. The target function for regression is a sine
> > > curve $y(x) = A \sin(\omega x)$. We report MSE errors with a 95% confidence interval.*
> > > |Method |Training |5-shot Testing| 10-shot Testing| 20-shot Testing|
> > > |:-------|:---------|:---------------|:----------------|:------------------|
> > > |MAML| 5-shot |1.13 ± 0.08|0.85 ± 0.14| 0.71 ± 0.12|
> > > |Meta-SGD| 5-shot |0.90 ± 0.16 |0.63 ± 0.12| 0.50 ± 0.10|
> > > |Meta-SGD+ALFA (reproduced)| 5-shot| 0.60 ± 0.04| 0.41 ± 0.02| 0.25 ± 0.22|
> > > |Meta-SGD+Meta-AdaM| 5-shot| **0.52** ± 0.04 | **0.35** ± 0.04|  **0.23** ± 0.01|
> > >
> > > |Method |Training |5-shot Testing| 10-shot Testing| 20-shot Testing|
> > > |:-------|:---------|:---------------|:----------------|:------------------|
> > > |MAML| 10-shot| 1.17 ± 0.16| 0.77 ± 0.11| 0.56 ± 0.08|
> > > |Meta-SGD| 10-shot| 0.88 ± 0.14| 0.53 ± 0.09| 0.35 ± 0.06|
> > > |Meta-SGD+ALFA (reproduced)| 10-shot| 0.72 ± 0.16| 0.45± 0.03| 0.26 ± 0.02|
> > > |Meta-SGD+Meta-AdaM| 10-shot| **0.66** ± 0.22| **0.34** ± 0.05| **0.23** ± 0.01|
> > >
> > >
> > > |Method |Training |5-shot Testing| 10-shot Testing| 20-shot Testing|
> > > |:-------|:---------|:---------------|:----------------|:------------------|
> > > |MAML| 20-shot |1.29 ± 0.20| 0.76 ± 0.12| 0.48 ± 0.08|
> > > |Meta-SGD| 20-shot| 1.01 ± 0.17| 0.54 ± 0.08| 0.31 ± 0.05|
> > > |Meta-SGD+ALFA (reproduced)| 20-shot| 1.01± 0.18 |0.48± 0.09| 0.25 ± 0.03|
> > > |Meta-SGD+Meta-AdaM| 20-shot| **0.95** ± 0.11| **0.43** ± 0.06| **0.23** ± 0.03|
> > > ___
> > > **Concern 1**:  *Provide results on regression tasks.*
> > > ___
> > > **Answer:**
> > >
> > > Thanks for your valuable suggestions. To further enhance our work, we conducted additional experiments on few-shot regression tasks. We follow the experiment setting of the Meta-SGD [2]. The regression task is to map the underlying function based on the input. The target function is $y(x) = A \sin(\omega x)$, where amplitude $A$, frequency $\omega$, and phase $b$ follow a uniform distribution with intervals [0.1, 5.0], [0.8, 1.2], and [0, $\pi$]. We train three few-shot regression learners on 5-shot, 10-shot, and 20-shot tasks. Each learner will test on 100 5-shot, 10-shot, and 20-shot testing tasks. We report MSE error with a 95% confidence interval.
> > >
> > > We compare MAML, Meta-SGD, Meta-SGD+ALFA[1], and Meta-SGD+Meta-AdaM. The result has shown that our method outperforms two major baselines, Meta-SGD and Meta-SGD+ALFA, in all settings. The results show that our proposed method is also effective on few-shot regression tasks.
> > > ___
> > > **Reference**:
> > >
> > > [1] S. Baik, M. Choi, J. Choi, H. Kim, and K. M. Lee. Meta-learning with adaptive hyperparameters,2020.
> > >
> > > [2] Z. Li, F. Zhou, F. Chen, and H. Li. Meta-SGD: Learning to learn quickly for few-shot learning,2017.2

---

> > > > ### Comment · Reviewer_7eNt · 2023-08-20
> > > >
> > > > Thanks to the authors for providing the regression results, I recommend they include them into their paper if space permits.
> > > >
> > > > I will keep my previously raised rating and recommend weak acceptance of this work.

---

> > > > > ### Author Response · Authors · 2023-08-21
> > > > >
> > > > > Thanks for your kind advice in your review. We will take your advice to further improve our final paper.

---

### Official Review · Reviewer_W1we · 2023-07-06

**Soundness:** 4 excellent
**Presentation:** 4 excellent
**Contribution:** 3 good
**Rating:** 7
**Confidence:** 5

**Summary:**

The paper proposes Meta-Adam, a meta-learned adaptive optimizer that employs momentum to rapidly adapt a meta-learned initial model to a few-shot task. The proposed framework consists of four components. First, an LSTM that predicts a learning rate for each weight inside the model using the weight update history in previous iterations. Furthermore, the optimizer employs momentum and additionally develops a look-ahead procedure for identifying environments where momentum hurts optimization and adjusting the parameters accordingly. Lastly, the loss function is weighted class-wise based on the class-specific loss during each gradient step to ensure that all class losses are optimized collectively. Experiments on mini-, tiered-ImageNet and Cifar100 benchmarks demonstrate the efficacy of the proposed method.

**Strengths:**

- The method proposed explores interesting ideas around better-optimizing learning rates and momentum in few-shot adaptive meta-learned models. The components of the method are well-motivated, derived to address specific problems, and empirically effective.
- Experiments demonstrate improvements in performance across various benchmarks compared to MAML-based benchmarks.
- Ablation studies provided demonstrate the empirical efficacy of each component of the proposed method and how much they contribute to the overall improvements in few-shot image classification accuracy.
- The paper is written clearly with reasonable notation and is easy to follow.

**Weaknesses:**

- All baselines that are compared are MAML-based. Although it's understandable given the focus of the paper on gradient adaptive few-shot image classification models, the method would still be compared against other groups (for instance, metric-based [1, 2, 3, 4]) when deciding to use a few-shot learner in an applied setting. It's useful to provide a discussion of this and potentially include a comparison to other non-MAML-based baselines.
- Figure 2 / Section 4.5 - it would be interesting to see what the variance bounds are for each optimization method as averages can be uninformative as to how much loss reduction can vary per task. Furthermore, ALFA seems to achieve much faster convergence although a worse optimum compared to Meta-AdaM. Why is that the case?
- 4.6 Dynamic class loss discussion lacks details as to the task/benchmark used in Figure 3. Without said details, it's difficult to evaluate whether the trends observed are statistically significant to support claims on reducing loss across all classes.

[1] Prototypical Networks for Few-shot Learning
[2] Improved Few-Shot Visual Classification
[3] Fast and Flexible Multi-Task Classification Using Conditional Neural Adaptive Processes
[4] Enhancing Few-Shot Image Classification with Unlabelled Examples

**Questions:**

See weaknesses for questions. Happy to improve my rating once the authors have addressed the concerns above.

**Limitations:**

There is no explicit discussion of limitations or potential negative societal impact of the work, although a brief discussion of some directions for future work, in particular with respect to negative learning rates, is provided. I strongly encourage the authors to provide a discussion of other potential limitations of the work.

---

> ### Author Rebuttal · Authors · 2023-08-09
>
> **Table 1:** Additional Comparison results using Convnet4 on Mini-ImageNet. We report performances in terms of accuracy (%) with standard deviation.
> |Dataset|Method| 5-way-1-shot|5-way-5-shot |
> |:-----------------:|:-------------------------:|:------------------------:|:------------------------:|
> |Mini-ImageNet|ProtoNet[1]|49.42 ± 0.78|68.20 ± 0.66|
> |Mini-ImageNet|Meta-AdaM(ours)|52.00 ± 0.49|70.70 ± 0.49|
> ___
>
> **Question 1**
>
> *All baselines that are compared are MAML-based. Although it's understandable given the focus of the paper on gradient adaptive few-shot image classification models, the method would still be compared against other groups (for instance, metric-based [1, 2, 3, 4]) when deciding to use a few-shot learner in an applied setting. It's useful to provide a discussion of this and potentially include a comparison to other non-MAML-based baselines.*
>
>
> **Answer**
>
> Thank you for highlighting these studies. In Table 1 above, we compare our result with ProtoNet [1] on Mini-Imagenet using Convnet4 as the backbone. The results show that our method outperforms the ProtoNet [1] in both settings.  For the other three methods, we want to note that the methods presented in [2,3,4] use pre-trained models. In contrast, our primary focus centers around the proposed Meta-Adam optimizer. Consequently, the performance results we have reported do not stem from pre-trained models, ensuring minimal interference from other factors. While time constraints have prevented us from providing performance data using pre-trained models, it's crucial to understand that our techniques complement them. As such, our methods can be integrated with gradient-based meta-learning strategies that employ pre-trained models as long as they employ an inner fine-tuning process.
> ___
>
> **Question 2:**
>
> *Figure 2 / Section 4.5 - it would be interesting to see what the variance bounds are for each optimization method, as averages can be uninformative as to how much loss reduction can vary per task. Furthermore, ALFA seems to achieve much faster convergence, although a worse optimum compared to Meta-AdaM. Why is that the case?*
>
> **Answer:**
>
> Thank you for your recommendation. In Figure 1 of the global rebuttal, we present the variance bounds of both our method and ALFA across 500 test tasks. From the figure, we can see that after the third step, the losses associated with the test tasks closely align with the mean loss. The variances after step 3 are around $1 \times 10^{−4}$. Therefore, the loss reduction works for the majority of
> tasks.
> For the convergence rate of the first several steps, we empirically show the predicted learning rates across test tasks in Figure 2 in the global rebuttal. The results show that our optimizer tends to predict a lower learning rate at the first few steps since the conflict between momentum and gradient is large. This can be reflected in Figure 3 in the global rebuttal, where momentum coefficients are smaller than
> gradients. Using a smaller learning rate will lead to a slow convergence rate in the first few steps, which provides spaces for accumulating high-quality momentum. Once the momentum quality is good after several steps, the predicted learning rates become larger and lead to a better optimum than ALFA. This also highlights our contribution that using double-look-ahead to handle highly variable momentum.
> ___
>
> **Question 3:**
>
> 4.6 Dynamic class loss discussion lacks details regarding the task/benchmark used in Figure 3. Without said details, it’s difficult to evaluate whether the trends observed are statistically significant to support claims on reducing loss across all classes.
>
> **Answer:**
>
> Thank you for drawing attention to this. In Section 4.6, we used a 5-way-1-shot task on the mini-Imagenet dataset. For the finalized version, we intend to broaden our evaluation to include additional tasks and benchmark datasets.
> ___
>
> **References:**
>
> [1] Snell, J., Swersky, K., & Zemel, R. (2017). Prototypical networks for few-shot learning. Advances in neural information processing systems
>
> [2] Bateni, P., Goyal, R., Masrani, V., Wood, F., & Sigal, L. (2020). Improved few-shot visual classification. In Proceedings of the IEEE/CVF Conference on Computer Vision and Pattern Recognition (pp. 14493-14502).
>
> [3] Requeima, J., Gordon, J., Bronskill, J., Nowozin, S., & Turner, R. E. (2019). Fast and flexible multi-task classification using conditional neural adaptive processes. Advances in Neural Information Processing Systems
>
> [4] Bateni, P., Barber, J., Van de Meent, J. W., & Wood, F. (2022). Enhancing few-shot image classification with unlabelled examples. In Proceedings of the IEEE/CVF Winter Conference on Applications of Computer Vision (pp. 2796-2805).

---

> > ### Comment · Reviewer_W1we · 2023-08-17
> >
> > The authors have adequately addressed the limitations noted in my original review. I strongly recommend that the authors include the discussion above within their camera ready and expand on the number of non-MAML baselines they compare to (including some that may have not been referenced explicitly). That being said, after reviewing the rebuttal, reviews by other reviewers, and the authors' responses, I am happy to recommend the paper for acceptance and will be improving my rating to full acceptance.

---

> > > ### Author Response · Authors · 2023-08-21
> > >
> > > Thanks for your kind advice in your review. We will take your advice to further improve our final paper.

---

### Official Review · Reviewer_cqvx · 2023-07-06

**Soundness:** 3 good
**Presentation:** 2 fair
**Contribution:** 3 good
**Rating:** 6
**Confidence:** 3

**Summary:**

This paper introduces a novel meta-learned optimizer with momentum for few-shot learning. The proposed approach incorporates the parameter-update history by leveraging LSTMs to adaptively adjust the learning rate. To enhance convergence speed,  the momentum is integrated into the meta-optimizer by using a double look-ahead mechanism. Furthermore, to address imbalanced learning, a dynamic class weighing schema is introduced, encouraging optimal model optimization by assigning more weight to influential classes. The effectiveness of the proposed method is evaluated on multiple benchmarks, showcasing its performance in few-shot learning scenarios.

**Strengths:**

* The proposed meta-optimizer is novel, as it considers the history of weight updates and momenta when adapting the learning rate, which distinguishes it from previous works in the field. However, as my research expertise lies outside of the meta-optimizer domain, I cannot ascertain the paper's coverage of previous literature. Thus, I eagerly anticipate feedback from other reviewers with expertise in this topic.

* The paper is written in a clear and concise manner, facilitating readers' comprehension of the ideas and technical contributions. Each sub-problem is precisely defined, including its limitations, followed by the proposal of techniques to address those issues. This clarity enhances the paper's accessibility and aids in conveying the research concepts effectively.

**Weaknesses:**

* The related work section may have missed some recent works or baseline methods on meta-optimizer for few-shot learning (FSL). All the existing works mentioned in Section 2.2.1 were published before 2020, which raises the question of whether there have been significant developments in this topic between 2021 and 2023. This could lead to confusion and a potential gap in the literature review.

* The writing in the paper lacks conciseness. The author repeats high-level ideas and the limitations of previous works using very similar expressions multiple times throughout the entire paper. For instance, the description provided in Line 118, Line 129, and Line 139 are identical, which can be redundant and could benefit from consolidation.

* The method section contains some content that should be included in the related works section. For example, in Line 145, the comparison between the proposed idea and a previous work should be presented in the related works section.

**Questions:**

* I would appreciate it if the authors and other reviewers could carefully verify the inclusion of all relevant previous works in this area within the paper. This thorough examination is crucial for evaluating the novelty of the paper, and it will significantly influence my final rating.

* I would expect the author could improve the conciseness of the writing.

* The dynamic loss contributes to the accuracy of the proposed method, but the paper does not mention such a technique in the introduction. I expect the author could provide an explanation.

**Limitations:**

The paper includes the limitation in section 4.8 but fails to mention the negative social impact.

---

> ### Author Rebuttal · Authors · 2023-08-09
>
> **Table 1:** Additional Comparison results using Convnet4 on Mini-ImageNet, TieredImageNet, and
> Cifar100 datasets. We report performances in terms of accuracy (%) with standard deviation.
>
> |     Dataset        |        Method                 |  5-way-1-shot  |  5-way-5-shot    |
> |:-----------------:|:-------------------------:|:------------------------:|:------------------------:|
> | Mini-Imagenet |     e3bm(2020)[1]        |53.2| 65.1
> | Mini-Imagenet | Sparse-MAML(2021)[4] |51.04 ± 0.59| 68.05 ± 0.84|
> |Mini-Imagenet |  MAML+SiMT (2022)[3| 51.49 ± 0.18| 68.74 ± 0.12|
> |Mini-Imagenet    |MeTAL(2022)[2]|52.62 ± 0.37 |70.52 ± 0.29|
> |Mini-Imagenet|Meta-AdaM (ours)| 52.00 ± 0.49| **70.70 ± 0.49**|
>
> |     Dataset        |        Method                 |  5-way-1-shot  |  5-way-5-shot    |
> |:-----------------:|:-------------------------:|:------------------------:|:------------------------:|
> |TieredImageNet |e3bm(2020)[1]| 52.1| 70.2|
> |TieredImageNet |Sparse-MAML(2021)[4]|56.39 ± 0.38|73.01 ± 0.24|
> |TieredImageNet |MAML+SiMT(2022)[3]|52.51 ± 0.21| 69.58 ± 0.11|
> |TieredImageNet |MeTAL(2022)[2]| 54.34 ± 0.31| 70.40 ± 0.21|
> |TieredImageNet |Meta-AdaM(ours)| 53.93 ± 0.49| **72.66** ± 0.49|
>
> |     Dataset        |        Method                 |  5-way-1-shot  |  5-way-5-shot    |
> |:-----------------:|:-------------------------:|:------------------------:|:------------------------:|
> |Cifar100|e3bm(2020)[1]| 39.9 |52.6|
> |Cifar100|Meta-AdaM (ours) |**41.11** ± 0.49| **56.32** ± 0.49|
> ___________________________________________________________
>
>
> Table 2: Additional Comparison results using Resnet12 on Mini-ImageNet and TieredImageNet. We
> report performances in terms of accuracy (%) with standard deviation.
> | Dataset| Method  | 5-way-1-shot|  5-way-5-shot |
> | :-----------------: | :-------------------------:| :------------------------:|:------------------------:|
> |Mini-ImageNet|MAML+SiMT(2022)[3]| 56.28 ± 0.63| 72.01 ± 0.21
> |Mini-ImageNet|MeTAL(2022)[2]| 59.64 ± 0.38 |76.20 ± 0.19|
> |Mini-ImageNet|Meta-AdaM(ours)| **59.89** ± 0.49| **77.92** ± 0.43|
>
> | Dataset| Method  | 5-way-1-shot|  5-way-5-shot |
> | :-----------------: | :-------------------------:| :------------------------:|:------------------------:|
> |TieredImageNet|MAML+SiMT(2022)[3]|59.72 ± 0.22 |74.40 ± 0.90|
> |TieredImageNet|MeTAL](2022)[2] |63.89 ± 0.43| 80.14 ± 0.40|
> |TieredImageNet|Meta-AdaM(ours)| **65.31** ± 0.48| **85.24** ± 0.35|
>
> ________________________________________________________________________________________________________________________
> **Weakness 1 & Question 1**:   *I would appreciate it if the authors and other reviewers could carefully verify the inclusion of all relevant previous works in this area within the paper. This thorough examination is crucial for evaluating the paper's novelty and will significantly influence my final rating.*
>
>
> **Answer**:
>
> Thanks for your suggestion. We have carefully examined the related works, and the novelty of our work is integrating momentum into a meta-learned adaptive optimizer. Existing methods cannot utilize momentum for weights update due to the high variances of momen at the initial update steps. We address this challenge by double look-ahead. Also, we propose to utilize momentum for adaptive learning rate prediction.
>
> To address your concern, we present the result of recent works in Table 1 and Table 2. The experiment section in our paper already presents sparse-MAML (2021)[4]. In addition, we include e3bm (2020)[1], MAML+SiMT (2022)[3], and MeTAL (2022)[2], which also work on improving the fine-tuning process. We can observe that our methods outperform the other methods in most of the settings.
> ________________________________________________________________________________________________________________________
>
> **Weakness 2 & Question 2**: *The writing in the paper lacks conciseness. The author repeats high-level ideas and the limitations of previous works using similar expressions multiple times throughout the paper. For instance, the description provided in Line 118, Line 129, and Line 139 are identical, which can be redundant and could benefit from consolidation.*
>
> **Answer**:
>
> Thanks for your suggestion. We will revise and reduce the redundancy in the final version.
>
> _________________________________________________________________________________________________________________________
> **Weakness 3**:  *The method section contains some content that should be included in the related works section. For example, in Line 145, the comparison between the proposed idea and previous work should be presented in the related works section.*
>
> **Answer**:
>
> Thanks for your suggestion. We will add the missing baseline in the related work section of our final paper.
>
> _____
>
> **Question 3**:  *The dynamic loss contributes to the accuracy of the proposed method, but the paper does not mention such a technique in the introduction. I expect the author could provide an explanation.*
>
> **Answer**:
>
> Thanks for pointing out this. We will add a discussion of dynamic loss in the introduction.
> _______
>
> **References**
>
> [1] Y. Liu, B. Schiele, and Q. Sun. An ensemble of epoch-wise empirical bayes for few-shot learning. In Computer Vision–ECCV 2020: 16th European Conference, Glasgow, UK, August 23–28, 2020, Proceedings, Part XVI 16, pages 404–421. Springer, 2020.29
>
> [2] H. K. D. C. J. M. K. M. L. Sungyong Baik, Janghoon Choi. Meta-learning with task-adaptive loss function for few-shot learning. In International Conference on Computer Vision (ICCV),2021
>
> [3] J. Tack, J. Park, H. Lee, J. Lee, and J. Shin. Meta-learning with self-improving momentum target. In Advances in Neural Information Processing Systems, 2022.34
>
> [4] J. von Oswald, D. Zhao, S. Kobayashi, S. Schug, M. Caccia, N. Zucchet, and J. Sacramento.
> Learning where to learn: Gradient sparsity in meta and continual learning, 2021.

---

> > ### Comment · Reviewer_cqvx · 2023-08-21
> >
> > Thanks for the author's response. Since most of my concerns are addressed, I would recommend an acceptance for the paper.

---

> > > ### Author Response · Authors · 2023-08-21
> > >
> > > Dear reviewer cqvx, we extend our sincere gratitude for the invaluable feedback you've provided. Your insights have played a pivotal role in enhancing the quality of our work.
> > >
> > > Upon reviewing your updated comments, we observed a notable shift towards a more positive perspective. In light of this, we wanted to kindly bring to your attention the possibility that the associated score might have been inadvertently overlooked for adjustment.
> > >
> > > Please know that we completely respect your decision. Once again, thank you for your invaluable feedback.

---

### Official Review · Reviewer_xuqU · 2023-07-09

**Soundness:** 2 fair
**Presentation:** 3 good
**Contribution:** 2 fair
**Rating:** 4
**Confidence:** 3

**Summary:**

In this paper, the authors aim to solve the few-shot learning problem by proposing a meta-learned learning rate learner. This learner utilizes weight-update history as input to predict more appropriate learning rates for rapid convergence. Furthermore, they incorporate momentum into the optimization process of few-shot learning via a double look-ahead mechanism, enabling rapid convergence similar to many-shot settings. Extensive experimental results are provided to show the effectiveness of the proposed method.

**Strengths:**

- This paper is well-written and easy to read. A lot of figures are included to explain the proposed method.

&nbsp;

- The proposed method is technically sound. It is reasonable to use meta-learning to adjust the learning rates.

&nbsp;

- Extensive experimental results are provided.


**Weaknesses:**

- The proposed idea of using a model to output the learning rates is not novel. A similar thing has been done in [a], which uses a meta-learned model to output a series of hyperparameters, including learning rates.

&nbsp;

- The performance of the proposed method is much lower compared to recent few-shot learning methods. For example, the 1-shot accuracy on miniImageNet is only 59.89% in Table 3. Existing methods, e.g., [b] and [c], achieve more than 68% accuracy in the same setting. It is not necessary that the proposed method beats all existing methods. However, it should not be much lower than other methods. At least, the performance should be comparable. Besides, it is necessary to indicate the proposed method doesn’t achieve the SOTA performance and compare the SOTA in the tables.

&nbsp;

- All important related works are missing, e.g., [a-d]. They should be compared and discussed.

&nbsp;

Overall, I think the quality of this paper is not satisfactory. The idea is a little bit incremental. The overall performance is not significant. Many important related works are missing. Therefore, I recommend rejection. The authors need to show the proposed method can be applied to recent popular baselines, e.g., [b, c, d]. I don’t think the current version is ready to be presented at NeurIPS.

&nbsp;

[a] An Ensemble of Epoch-wise Empirical Bayes for Few-shot Learning, ECCV 2020.

[b] Partner-Assisted Learning for Few-Shot Image Classification, ICCV 2021.

[c] DeepEMD: Differentiable Earth Mover’s Distance for Few-Shot Learning, TPAMI.

[d] Rectifying the Shortcut Learning of Background for Few-Shot Learning, NeurIPS 2021.


**Questions:**

Please address my concerns in the "weaknesses"

**Limitations:**

See my concerns in the "weaknesses"

---

> ### Author Rebuttal · Authors · 2023-08-09
>
> **Table 1:** Additional Comparison results using Convnet4 on Mini-ImageNet, TieredImageNet, and
> Cifar100 datasets. We report performances in terms of accuracy (%) with standard deviation.
>
> |Dataset|Method| 5-way-1-shot|5-way-5-shot |
> |:-----------------:|:-------------------------:|:------------------------:|:------------------------:|
> | Mini-Imagenet |     e3bm[3]        |53.2| 65.1|
> |Mini-Imagenet |Meta-AdaM (ours)| **52.00** ± 0.49| **70.70** ± 0.49|
> |TieredImageNet |e3bm[3]| 52.1| 70.2|
> |TieredImageNet |Meta-AdaM(ours)| **53.93** ± 0.49| **72.66** ± 0.49|
> |Cifar100|e3bm[3]| 39.9 |52.6|
> |Cifar100|Meta-AdaM (ours) |**41.11** ± 0.49| **56.32** ± 0.49|
> ________________________________________________________________________________________________________________________
> **Question 1**: *The proposed idea of using a model to output the learning rates is not novel. A similar thing has been done in [a], which uses a meta-learned model to output a series of hyper-parameters, including learning rates.*
>
> **Answer**:
>
>  It seems there’s been a misunderstanding regarding the originality of our methods. We wish to emphasize our pioneering contributions in two aspects:
> * Integrating momentum into a meta-learned adaptive optimizer.
> * Utilizing the history of weight changes for adaptive learning rate prediction
>
>  Existing methods have been hindered from employing momentum for weight updates due to pronounced variances in momentum during initial optimization steps. To overcome this limitation, we introduce a double-look-ahead strategy. Furthermore, our approach involves leveraging momentum for adaptive learning rate prediction, which more accurately reflects weight change compared to other adaptive learning rate prediction methods. The key difference between our work and [1] and other related works is that our learning rate prediction is based on the update history. Our LSTM network takes accumulated momentum and current gradients to grasp the weight update history, which is more critical than gradients and weights [1, 2].
> In contrast, [3 ] and other related works mostly use average loss, input features or weight, and current gradient as their inputs to determine the learning rate. They focus more on the current update rather than the update history. The experiment studies in section 4.2 and section 4.3 and the additional results presented in Table 1 show that our methods are more effective than other related works.
>
> __________________________________________________________________________________________________________________________
> **Question 2**: *The performance of the proposed method is much lower compared to recent few-shot learning methods. For example, the 1-shot accuracy on miniImageNet is only 59.89 % in Table 3. Existing methods, e.g., [b] and [c], achieve more than 68% accuracy in the same setting. It is not necessary that the proposed method beats all existing methods. However, it should not be much lower than other methods. At least, the performance should be comparable. Besides, it is necessary to indicate the proposed method doesn’t achieve the SOTA performance and compare the SOTA in the tables.*
>
> **Answer**:
>
> Thank you for highlighting these studies. We want to note that the methods presented in [4,5,6] (b,c,d in your review) use pre-trained models. In contrast, our primary focus centers around the proposed Meta-Adam optimizer. Consequently, the reported performance results do not stem from pre-trained models, ensuring minimal interference from other factors. While time constraints have prevented us from providing performance data using pre-trained models, it’s crucial to understand that our techniques complement them. As such, our methods can seamlessly be integrated with gradient-based meta-learning strategies that employ pre-trained models.
>
> _________________________________________________________________________________________________________________________
> References
>
> [1] I. Goodfellow, Y. Bengio, and A. Courville. Deep learning. MIT Press, 2016.
>
> [2] M. J. Kochenderfer and T. A. Wheeler. Algorithms for optimization. MIT Press, 2019.
>
> [3] Y. Liu, B. Schiele, and Q. Sun. An ensemble of epoch-wise empirical bayes for few-shot learning.In Computer Vision–ECCV 2020: 16th European Conference, Glasgow, UK, August 23–28, 2020, Proceedings, Part XVI 16, pages 404–421. Springer, 2020.
>
> [4] X. Luo, L. Wei, L. Wen, J. Yang, L. Xie, Z. Xu, and Q. Tian. Rectifying the shortcut learning of background for few-shot learning. Advances in Neural Information Processing Systems, 34:13073–13085, 2021.
>
> [5] J. Ma, H. Xie, G. Han, S.-F. Chang, A. Galstyan, and W. Abd-Almageed. Partner-assisted learning for few-shot image classification. In Proceedings of the IEEE/CVF International Conference on Computer Vision, pages 10573–10582, 2021
>
> [6] C. Zhang, Y. Cai, G. Lin, and C. Shen. Deepemd: Differentiable earth mover’s distance for few-shot learning. IEEE Transactions on Pattern Analysis and Machine Intelligence, 45(5):5632–5648, 2022.50

---

> > ### Comment · Reviewer_xuqU · 2023-08-10
> >
> > Thanks for the response from the authors.
> >
> > **For Q1**: The authors didn't really answer my question. I agree that the authors propose a better way to meta-learn the learning rates for few-shot learning. However, the basic idea of "meta-learning the learning rates" is not novel. Therefore, the overall contribution is this paper is somewhat incremental and limited. In the authors' response, they only emphasize the technical detail differences between the proposed method and existing works.
> >
> > **For Q2**: Again, the authors only explained their experimental settings and didn't provide any solid results.
> >
> > I will consider upgrading my rating if the authors can:
> > - Change the claims in the paper and highlight that "meta-learning the learning rates" has been explored a lot in existing works, e.g., [a] and [e].
> > - Provide the results using pre-trained models, and show their method still works better.
> >
> > [a] An Ensemble of Epoch-wise Empirical Bayes for Few-shot Learning, ECCV 2020.
> >
> > [e] Meta-SGD: Learning to Learn Quickly for Few-Shot Learning. https://arxiv.org/abs/1707.09835

---

> > > ### Author Response · Authors · 2023-08-21
> > > **Result on pretrained network**
> > >
> > > |Dataset| Method| 5-way-5-shot|
> > > |:---------|:-----------|:-----------------|
> > > |Mini-ImageNet|AM3+TRAML[4]|79.54 ± 0.60|
> > > |Mini-ImageNet|baseline[1]| 81.38 ± 0.41|
> > > |Mini-ImageNet|Net-Cosine[5]|81.57 ± 0.56|
> > > |Mini-ImageNet|MABAS[3]|82.70 ± 0.54|
> > > |Mini-ImageNet|IEPT[8]| 82.90 ± 0.30|
> > > |Mini-ImageNet|MELR [2]| 83.40 ± 0.28|
> > > |Mini-ImageNet|DeepEMD [7]| 83.47 ± 0.61|
> > > |Mini-ImageNet|COSCO [6] |85.16 ± 0.42|
> > > |Mini-ImageNet|Meta-AdaM + pretrained (ours)| 84.46 ± 0.36|
> > > ___
> > > Sorry for the late response. We have been working on getting results for combining our Meta-AdaM with the pre-trained model ResNet12.
> > > ___
> > > **Concern 1:**  *Change the claims in the paper and highlight that "meta-learning the learning rates" has been explored a lot in existing works, e.g., [a] and [e].*
> > >
> > > **Answer:**
> > > Thank you for your great suggestion. We will revise the paper’s claims and place more emphasis on prior research regarding learning rate meta-learning in both the Introduction and Related Work sections of the final version.
> > > ___
> > >
> > > **Concern 2:** *Provide the results using pre-trained models, and show their method still works better.*
> > >
> > > **Answer:**
> > > To address the concern, we conducted further experiments utilizing pre-trained networks for the 5-way-5-shot task on the Mini-ImageNet dataset. Table 1 summarizes our findings in comparison to current state-of-the-art models. We employed a pre-trained ResNet12 as the backbone network, as provided by [ 6]. Our approach yields results in line with top-performing methods, and notably, our
> > > result is close to that of COSCO [6], which leverages additional data, such as object positioning.
> > >
> > > ___
> > > **References:**
> > > [1] W.-Y. Chen, Y.-C. Liu, Z. Kira, Y.-C. F. Wang, and J.-B. Huang. A closer look at few-shot classification. arXiv preprint arXiv:1904.04232, 2019.
> > >
> > > [2] N. Fei, Z. Lu, T. Xiang, and S. Huang. Melr: Meta-learning via modeling episode-level relationships for few-shot learning. In International Conference on Learning Representations,2020
> > >
> > > [3] J. Kim, H. Kim, and G. Kim. Model-agnostic boundary-adversarial sampling for test-time generalization in few-shot learning. In Computer Vision–ECCV 2020: 16th European Conference, Glasgow, UK, August 23–28, 2020, Proceedings, Part I 16, pages 599–617. Springer, 2020.
> > >
> > > [4] A. Li, W. Huang, X. Lan, J. Feng, Z. Li, and L. Wang. Boosting few-shot learning with adaptive margin loss. In Proceedings of the IEEE/CVF conference on computer vision and pattern recognition, pages 12576–12584, 2020.
> > >
> > > [5] B. Liu, Y. Cao, Y. Lin, Q. Li, Z. Zhang, M. Long, and H. Hu. Negative margin matters: Understanding margin in few-shot classification. In Computer Vision–ECCV 2020: 16th European Conference, Glasgow, UK, August 23–28, 2020, Proceedings, Part IV 16, pages 438–455. Springer, 2020
> > >
> > > [6] X. Luo, L. Wei, L. Wen, J. Yang, L. Xie, Z. Xu, and Q. Tian. Rectifying the shortcut learning of background for few-shot learning. Advances in Neural Information Processing Systems, 34:13073–13085, 2021
> > >
> > > [7] C. Zhang, Y. Cai, G. Lin, and C. Shen. Deepemd: Differentiable earth mover’s distance for few-shot learning. IEEE Transactions on Pattern Analysis and Machine Intelligence, 45(5):5632–5648, 2022.
> > >
> > > [8] M. Zhang, J. Zhang, Z. Lu, T. Xiang, M. Ding, and S. Huang. Iept: Instance-level and episode-level pretext tasks for few-shot learning. In International Conference on Learning Representations, 2020.

---

### Author Rebuttal · Authors · 2023-08-09

Dear Reviewers,

Firstly, we'd like to express our deep gratitude for the comprehensive review and insightful feedback on our paper. We have carefully addressed each of your comments, and our individual responses to each query can be found in the subsequent sections.

Incorporating your feedback, we've endeavored to enhance the overall clarity and substance of our paper. To this end, we've conducted supplementary experiments for dynamic class weighting, the predicted learning rates, and computed coefficients for momentum. Some figures are now integrated into the attached PDF.

Your diligent review and valuable suggestions have been important in improving the quality of our manuscript. We deeply value your expertise and the time you invested in assessing our work.

Best regards,

The Authors

---

### Decision · Program_Chairs · 2023-09-21

**Decision:**

Accept (poster)

**Comment:**

This paper introduces a novel meta-learned optimizer with momentum for few-shot learning. The majority of reviewers awarded positive scores, recognizing the innovation and robust technical contribution of this study. This positive assessment was particularly reinforced after the authors provided additional technical details and experimental validation in their rebuttal. Consequently, the AC concurs with the majority of reviewers in recommending the acceptance of this manuscript.